# Single-cell transcriptomics reveals FXR1 as an actionable target for siRNA therapy in ovarian cancer

Jasmine George[1], Xiaolong Ma [2,6], Ishaque Pulikkal Kadamberi[1,6], Ajay Nair[3,6], Sonam Mittal[1], Elaheh Hashemi[4], Sudhir Kumar[1], Mona Singh[1], Anjali Geethadevi[1], Meenakshi Pradeep[1], Anupama Nair[1], Shirng-Wern Tsaih [1], Julie M. Jorns[5], Subramaniam Malarkannan[4], Felix Dietlein [3], Chien-Wei Lin [2]✉, Sunila Pradeep [1]✉ & Pradeep Chaluvally-Raghavan [1]✉

Ovarian cancer is one of the leading causes of cancer-related mortality among women and remains exceptionally difficult to manage and treat effectively in the clinic. Fragile X-related protein 1 (FXR1) is highly amplified and over-expressed in ovarian and several other cancers. FXR1 is a key regulator of the translation of multiple oncogenes and therefore represents a vulnerable target for cancer therapy. RNA interference (RNAi) of FXR1 using a locked nucleic acid (LNA) form of siRNA (siFXR1-LNA) inhibits tumor growth, ascites formation, and metastasis of ovarian cancer more efficiently than the native form of FXR1 siRNA in vivo. LNA modification of siRNA improves resistance to RNase mediated degradation and enhances tumor tissue uptake of siRNA with robust inhibition of target mRNA in tumor tissues. Single-cell RNA sequencing (scRNA-seq) analysis of ascites composed of tumor, stromal, and immune cells analysis reveals that FXR1 silencing suppresses tumor cell proliferation and reduces tumor-promoting M2-like macrophages. FXR1 silencing also increases cytotoxic T cells, NK cells, and dendritic cells with anti-tumor characteristics in vivo. Collectively, our data establishes FXR1 as an important regulator of oncogenic processes in cancer tissues and serves as a therapeutic liability. Therefore, FXR1 silencing in tumor tissues provides an effective strategy to treat tumors expressing high levels of FXR1.

Ovarian cancer, a leading cause of cancer deaths among women with a 1 in 87 lifetime risk, often goes undetected until its aggressive later stages, leading to low survival rates and frequent recurrences despite early treatment success. Studies from The Cancer Genome Atlas[1], us[2], and others[3] have reported that ovarian cancer is driven by copy number variations (CNVs), such as copy number gain, amplification, or deletion of the genome. Many genes, such as *PI3KCA*[4], *EVI1*[5], and *TERC*[6] were reported as amplified within the 3q26 locus amplification for their role in ovarian cancer progression. We have reported that non-coding microRNAs, such as the miR569 and miR551b, that are amplified as part of the 3q26.2 locus, contribute to the oncogenesis and progression of breast and ovarian cancers[2,7,8]. Although many genes have been reported for ovarian cancer progression as driver genes or passenger genes, targeted therapy is not well developed for ovarian cancer.

[1]Department of Obstetrics and Gynecology, Medical College of Wisconsin, Milwaukee, WI, USA. [2]Division of Biostatistics, Data Science Institute, Medical College of Wisconsin, Milwaukee, WI, USA. [3]Computational Health Informatics Program, Boston Children's Hospital, Harvard Medical School, Boston, MA, USA. [4]Versiti Blood Research Institute, Milwaukee, WI, USA. [5]Department of Pathology, Medical College of Wisconsin, Milwaukee, WI, USA. [6]These authors contributed equally: Xiaolong Ma, Ishaque Pulikkal Kadamberi, Ajay Nair. ✉e-mail: chlin@mcw.edu; spradeep@mcw.edu; pchaluvally@mcw.edu

Our recent work identified that fragile X-related protein 1 (FXR1), amplified within the 3q26 genomic region, is a master regulator of stability and translation of several oncogenic mRNAs[9]. FXR1 is copy-gained or overexpressed in many cancers, such as those of lung, ovary, esophagus, head and neck, cervical, and uterine[9]. Therefore, silencing FXR1 in tumor tissues could offer therapeutic benefits for ovarian and multiple other cancers that exhibit FXR1 amplification and/or over-expression. However, there is no inhibitor readily available that targets and inhibits FXR1 for therapy. Hence, the aim of this study is to develop RNA interference (RNAi) techniques, employing both in vitro and in vivo models, to target and suppress FXR1 for cancer therapy.

RNAi was a Noble Prize-winning discovery made by Dr. Craig Mello and Andrew Fire in 1998[10,11]. RNAi based therapeutic approaches haven't received any significant attention for clinical care until 2018 when FDA approved the first RNAi drug Patisiran to treat hereditary transthyretin amyloidosis, and subsequently RNAi drugs Givosiran and Lumasiran got approved for treating hepatic porphyria and hyperox-aluria in 2019 and 2020, respectively[12–14]. Since then, RNA-based ther-apy has become an important area of research for silencing many genes which were considered undruggable previously. However, the therapeutic potential of RNAi has not met the goal of treating diseases like cancers as expected, potentially due to lack of cancer cell specific RNAi targets and/or effective delivery mechanisms. Additionally, the impact of RNAi on normal cells and immune cell toxicity has not been thoroughly addressed for cancer therapy. Given this background, our research aims to explore the potential of using FXR1 siRNA not only for its ability to inhibit cancer cell growth and induce the death of cancer cells, but also for its immune-supporting effects, which could enhance anti-cancer responses.

Several challenges currently limit the clinical use of siRNA-based therapies, including difficulties in delivering siRNA specifically to tar-get cells, low uptake levels within these cells, and rapid degradation of siRNA by exonucleases and RNases. Another major concern is the off-target effects of siRNA, which silence unintended RNA targets, potentially leading to adverse outcomes[15]. Introducing chemical modifications in siRNAs evolved as a strategy to improve the stability of small RNAs by reducing the impact of nuclease induced degradation[16,17]. The chemical modifications can be introduced to the 5'- or 3'- terminus, backbone, sugar, or in the nucleotide bases of siRNAs[17,18]. Among the chemical modifications, phosphorothioates (PS), 2'-O-methylation (2'-O-Me), 2'-O-allyl, 2'-deoxy-fluorouridine and locked nucleic acid (LNA) forms were known for improving the stabi-lity of siRNA for therapy. LNA modification is a bicyclic nucleic acid where a ribonucleoside monomer is linked between the 2'-oxygen and the 4'-carbon atoms with a methylene unit[19]. In contrast to other modifications, the LNA modifications have been reported to sub-stantially increase the potency and activity of siRNAs in both in vitro and in vivo studies with minimal toxicity[20–23]. Another challenge in RNAi research is the development of efficient and safe delivery sys-tems. Recent advances, such as GalNAc conjugates and lipid nano-particles (LNPs), have emerged as effective strategies for encapsulating siRNAs in ionizable lipids for delivering them intra-cellularly for target gene knockdown[24,25].

Although many RNAi studies have reported anti-tumor effects of siRNAs in vitro and in vivo, not many of them were advanced through clinical trials for cancer. This drawback was mainly due to the non-specific targeting of siRNAs on other RNAs and their effects on non-tumor cells. Therefore, it is required to validate the effects of siRNAs in whole tumor tissues comprehensively using advanced technologies like single-cell RNA sequencing (scRNA-seq). However, scRNA-seq has not yet been fully utilized in the current drug discovery approaches, including siRNA therapy to evaluate the effects of drugs broadly in all cell populations in the tumor tissues.

Ovarian cancer is a peritoneally progressing cancer where the tumor cells shed into the peritoneal cavity, then circulates through ascites fluid and undergo transcoelomic metastasis a.k.a. peritoneal seeding[26,27]. Consequently, tumor cell clusters and spheroids isolated from ascites are considered as a representative model of studying transcriptomic signatures of advanced-stage ovarian cancer[28]. Given the nature of ovarian cancer, high throughput studies have utilized these ascites-derived clusters and spheroids to perform tran-scriptomic analyses at the single-cell level, providing insights into tumor heterogeneity and microenvironmental interactions[29–31].

In this work, we demonstrate that the in vivo delivery of LNA-modified siRNAs targeting FXR1 markedly suppresses ovarian cancer growth and metastasis in both immunocompetent and immunocom-promised mouse models. scRNA-seq of ascites from siFXR1-LNA–treated tumor-bearing mice reveals that FXR1 silencing restricts tumor cell expansion while promoting the enrichment of M1-like macrophages and cytotoxic Cd8$^+$ T cells, thereby enhancing anti-tumor immune responses. Collectively, these findings provide mechanistic insight into how FXR1-directed RNA interference inhibits ovarian cancer growth and progression in vivo and support FXR1 as a therapeutic vulnerability in ovarian cancer.

## Results

### Sequence specific screening identifies highly potent siRNAs that enhance FXR1 silencing, inhibit cell growth, and promote apoptosis

siRNAs bind to complementary mRNA sequences to inhibit gene expression. Thus, each target sequence is expected to exhibit varying levels of gene silencing. Therefore, we designed five siRNAs numbered as seq1 to seq5 (Table S1), which specifically target coding sequences (CDS) or the sequences in untranslated region (UTR) of FXR1 gene as marked in Supplementary Fig. 1a. First, we used siRNAs (siFXR1) at two different concentrations 2.5 nM and 5 nM and the respective scram-bled control siRNA (siCont) at 5 nM and found that the transfection of seq2 and seq3 siRNAs, targeting the CDS exhibited ~70−85% inhibition of FXR1 protein in ovarian cancer cell lines (Fig. 1a). While we observed ~80% inhibition of FXR1 protein by seq5 in HeyA8 and OVCAR8 cells, the level of inhibition was only ~50% in the Kuramochi cell line. In contrast, the seq1 and seq4 sequences exhibited low levels of FXR1 inhibition relative to the other sequences (Fig. 1a). In brief, we observed that CDS-targeting siRNAs demonstrated superior level of FXR1 inhibition compared to the non-CDS targeting regions (Supple-mentary Fig. 1a, a).

Next, we determined the knockdown effects of all five siRNAs at their lowest effective concentration of 2.5 nM on cell cycle phases using flow cytometry. Similar to the sequence specific effects of siRNAs on FXR1 inhibition (Fig. 1a), seq2 and seq3 exhibited the most profound effect on G1 phase arrest and a reduction in S phase (Supplementary Fig. 1b). Similarly, those sequences inhibited G1-phase proteins CDK2, CDK4, CDK6, Cyclin D1 and Cyclin E1 levels, most effectively compared to the other sequences (Supplementary Fig. 1c). In conjunction, FXR1 siRNAs seq2 and seq3 induced ~70% cellular apoptosis, accompanied by reduced BCL2 and increased PARP cleavage compared to siCont (Supplementary Fig. 1c, d). In agreement with the level of FXR1 knockdown, a modest increase in apoptosis was only observed with seq1, seq4, and seq5 (Supplementary Fig. 1d). We and others have reported that cMYC is a key target of FXR1[9,32]. Supporting this notion, cMYC protein level was also inhibited mostly by seq2 and seq3 siRNAs (Supplementary Fig. 1c). Similarly, seq2 and seq3 siRNAs also inhibi-ted colony forming ability of ovarian cancer cells (Supplemen-tary Fig. 1e).

### FXR1 siRNA preferentially inhibits the growth of FXR1-high expressing cancer cells compared to normal cells

One of the key challenges of siRNA therapy for cancer is its uptake by normal cells, leading to both targeted and non-specific effects. Therefore, it is crucial to assess the growth-inhibitory effects of

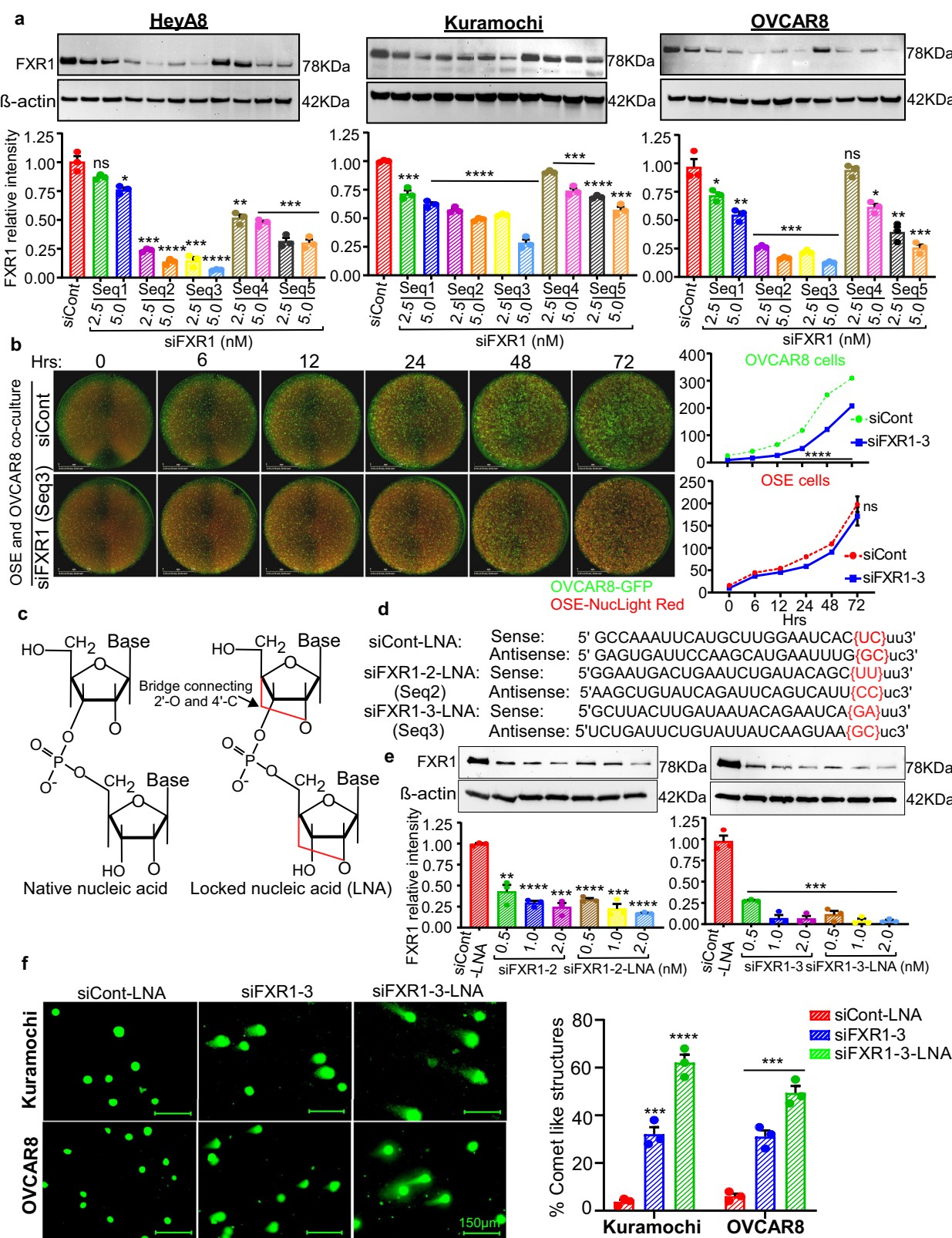

FXR1 siRNA specifically in cancer cells, ensuring minimal harm to non-tumor cells, along with immune cell activation. To address this, we selected the most potent FXR1 siRNAs (seq2 and seq3) and assessed their effects on tumor growth and apoptosis by transfecting normal and cancer cells either independently or in co-culture. In conjunction with FXR1 mRNA expression data (Supplementary Fig. 2a), the transfection of FXR1 siRNA effectively inhibited the growth of tumor cells, which express high levels of FXR1. In contrast, the normal cells that express low levels of FXR1 expression were unaffected by FXR1 siRNA. This data suggests that the growth-inhibitory effects caused by seq2 and seq3 siRNAs are primarily driven through target-specific inhibition of FXR1 (Supplementary Fig. 2b). Next, we employed

**Fig. 1 | LNA modification enhances knockdown efficiency, tumor penetration, and antitumor activity of FXR1 siRNAs. a** Representative Western blot image (top) shows FXR1 level in ovarian cancer cells were transfected with 5 nM siCont and 2.5 nM or 5 nM siRNAs (seq1 to 5) for FXR1 for 48 h. FXR1 levels were quantitated using β-actin loading control (bottom). Data represent mean ± SEM of three biological replicates. Significance was determined by unpaired two-tailed Student's t-test (*p < 0.05, **p < 0.01, ***p < 0.001, ****p < 0.0001, ns, non-significant). Uncropped blots are provided in Source Data file 1. **b** GFP + OVCAR8 cells co-cultured with NucLight Red-labeled OSE cells for 24 h, then transfected with siFXR1 (seq3) or siCont at 2.5 nM. Images were captured every 6 h for 3 days using the IncuCyte live-cell analysis system, and representative images were presented (left). Scale bars represent 2 mm. The graph displays time-dependent cell counts of GFP + OVCAR8 cells and NucLight Red-labeled OSE cells that were transfected with FXR1 siRNAs (right) (n = 9, represents nine biologically independent samples). Significance was determined by two-way ANOVA followed by Sidak's multiple comparisons test (****p < 0.0001, ns, non-significant). **c** Chemical structures of locked nucleic acid (LNA) modified form of siRNAs. The ribose ring is connected by

a methylene bridge between the 2'-O and 4'-C atoms, "locking" the ribose ring. **d** Oligonucleotide sequences of siCont-LNA and siFXR1-LNAs (seq2 and seq3) target FXR1, where red letters indicate LNA modified nucleotides at the 3'end of sense and antisense strands. **e** Representative Western blot images show FXR1 level in ovarian cancer cells were transfected with siFXR1-2, siFXR1-2-LNA, siFXR1-3 or siFXR1-3-LNA for 48 h (top). FXR1 levels were quantified using β-actin loading control (bottom). Data represent mean ± SEM of three biological replicates. Significance was determined by unpaired two-tailed Student's t-test (**p < 0.01, ***p < 0.001, ****p < 0.0001). Uncropped blots are provided in Source data file 1. **f** Kuramochi and OVCAR8 cells were transfected with siCont-LNA, siFXR1-3, or siFXR1-3-LNA. DNA damage was detected by COMET assay after 48 h of siRNA transfections. Representative images captured from five random fields of comet-like structures (left), and % of comet-like structures was quantified (right) (n = 3, represents three biologically independent samples). Scale bars represent 150 μm. Data represent mean ± SEM of three biological replicates. Significance was determined by unpaired two-tailed Student's t-test (***p < 0.001, ****p < 0.0001). Exact p-values for (**a, b, e, f**) are included in the Source data file 2.

a co-culture model where GFP-labelled OVCAR8 cells expressing high levels of FXR1 co-cultured with RFP-labeled normal ovarian surface epithelial cells (OSE) or fallopian tube epithelial cells (FTE) expressing very low levels of FXR1 in a single well. As expected, treatment with FXR1 siRNA reduced the overall GFP signal, indicating inhibition of tumor cell growth, while the unchanged RFP signal reflected that non-tumor cells expressing low levels of FXR1 were not affected by FXR1 siRNA treatment (Fig. 1b and Supplementary Fig. 2a, c).

We also conducted a second co-culture assay, where GFP-labelled OVCAR8 cells were incubated with normal OSE or FTE cells. Upon reaching approximately 80% confluence, the OVCAR8-GFP/OSE and OVCAR8-GFP/FTE co-cultures were transfected with either siFXR1 (seq3) or the corresponding siCont. GFP signal was then quantified at 2-day intervals for up to 6 days to assess tumor cell growth. This assay also demonstrated a decrease in the overall percentage of GFP signal from OVCAR8 cells without affecting the viability of non-FXR1 expressing normal OSE cells (Supplementary Fig. 2d).

### LNA modified siRNA improve FXR1 silencing and enhance cancer cell death compared to native siRNA

Based on the reports that LNA modifications can improve gene silencing potency with minimal toxicity[20–23], we introduced LNA modification at the 3'site in both seq2 and seq3 (hereafter referred to as siFXR1-2-LNA and siFXR1-3-LNA, respectively) (Fig. 1c, d). Next, we evaluated whether this modification enhanced their growth-inhibitory effect in cancer cells. Remarkably, siFXR1-3-LNA achieved over 90% FXR1 knockdown at just 0.5 nM, whereas such effect was obtained with 2 nM of the unmodified (native) siRNA (Fig. 1e). Compared to siFXR1-3-LNA, siFXR1-2-LNA was not able to exert such superior activity in knocking down FXR1 in its low concentrations.

In conjunction, siFXR1-3-LNA was again able to cause G1 arrest more prominently and reduced G2/M phase than native form of siFXR1-3 in both Kuramochi and OVCAR8 cells (Supplementary Fig. 3a). Proteins associated with cell cycle regulators such as Cyclin D1, Cyclin E1, CDK2, CDK4, CDK6, cMYC, and Bcl-2 were also inhibited predominantly by siFXR1-3-LNA than its native form. In contrast, the inhibition induced by siFXR1-2-LNA was less effective in comparison with siFXR1-3-LNA on inhibiting above cell cycle regulators (Supplementary Fig. 3b). Similarly, a subsequent increase was observed in the levels of cleaved PARP, caspase 3/7 activity, and cellular apoptosis when OVCAR8 and Kuramochi cancer cells were transfected with LNA-modified siFXR1-3 compared to its native form (Supplementary Fig. 3b−e). Our comet assay[33,34] also demonstrated a markable increase in DNA damage potentially due to cellular apoptosis in ovarian cancer cells, particularly when the cells were transfected with siFXR1-3-LNA compared to the native siRNA (Fig. 1f).

### LNA modified FXR1 siRNA demonstrates superior cellular and tissue uptake with improved stability over nucleases

The primary objective of this study is to assess the therapeutic effects of LNA-modified FXR1 siRNA on tumor growth inhibition via FXR1 silencing in preclinical models. Therefore, we sought to use an intermediary approach of 3D culture model that connects traditional 2D cultures and in vivo systems. Ovarian cancer cells were cultured on an extracellular matrix to form tumor spheroids, allowing us to simulate in vivo tumor growth and assess the growth-inhibitory effects of FXR1 siRNA more accurately. Similar to its effect in 2D culture (Fig. 1), siFXR1-LNA was able to inhibit the growth of ovarian cancer spheroids more efficiently than native siRNA (Fig. 2a). Furthermore, high level uptake of siFXR1-LNA was also observed by tumor spheroids compared to native siRNA (Fig. 2b and Supplementary Movies 1 and 2).

Once internalized, siRNA is often trapped in the endosome and transported to the lysosome, where it gets degraded by lysosomal enzymes quickly, significantly reducing its effectiveness[35,36]. Therefore, we decided to determine whether LNA modification can rescue siRNA from lysosomal degradation. Herein, we used Texas red labeled siRNAs of both native and LNA forms and transfected into cancer cells and the cellular uptake levels were determined by imaging (Fig. 2c). Strikingly, we observed a high level of siFXR1-LNAs uptake within 1 h after transfection in two different cancer cell lines Kuramochi and OVCAR8, whereas only modest level of siRNA uptake was observed when cells were transfected with native siRNA (Fig. 2c and Supplementary Fig. 4a). Consistently, siFXR1-LNAs showed minimum localization within lysosomes, suggesting reduced lysosomal degradation. In contrast, native siRNA displayed a strong lysosomal accumulation compared to LNA form (Fig. 2c and Supplementary Fig. 4a). This data suggests that LNA modified siRNA exhibits greater stability and can evade lysosome mediated degradation. The terminal deoxynucleotidyl transferase dUTP nick end labeling (TUNEL) assay also showed an increase in the level of cellular apoptosis when siFXR1-LNA transfected cells compared to those transfected with native siRNA (Fig. 2d and Supplementary Fig. 4b). These findings led us to investigate whether LNA modification protects siRNA from RNase activity and if encapsulating LNA form siRNA in nanoparticles further enhances its stability. Here, we used commercially available polyethylenimine (PEI) nanoparticle named jetPEI for encapsulation of siRNAs and found that unmodified siRNA, either in free or in encapsulated form, was degraded rapidly compared to LNA-modified siRNA (Fig. 2e). Notably, jetPEI encapsulation of siFXR1-LNA enhanced its stability and demonstrated least degradation in the presence of RNase (Fig. 2e).

We also used serum as a physiologically relevant source of exonucleases and RNases (ribonucleases) to assess whether LNA modification and PEI encapsulation protects siRNA from RNases mediated

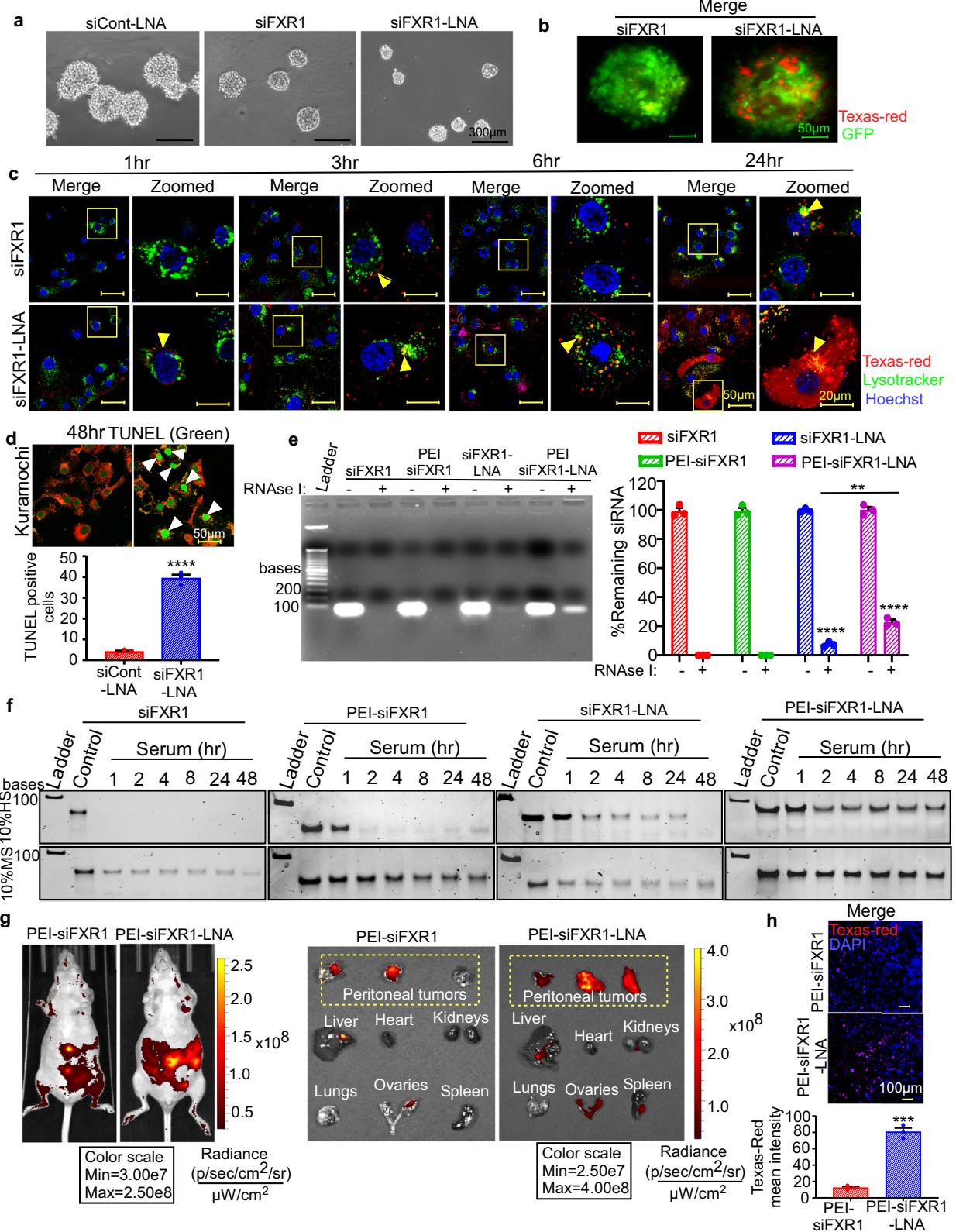

degradation. Consistent with the earlier results, LNA modifications in siRNAs improved stability over native siRNAs, and this stability was further enhanced upon encapsulation in jetPEI nanoparticles (Fig. 2f). Next, we examined whether siFXR1-LNA exhibits improved uptake in tumor tissues compared to normal organs when injected into tumor-bearing mice. As anticipated, siFXR1-LNA showed enhanced tissue uptake in peritoneal tumors and tumors at organ sites, with minimal

uptake in tissues devoid of tumor growth (Fig. 2g). Specifically, a high level of siRNA uptake was observed in tumor tissues and ovaries with tumor presence, whereas no uptake was detected in lungs and heart, which were free of tumor growth (Fig. 2g). Uptake was also very minimal in the liver, kidneys, and spleen, where a minimal tumor presence was observed (Fig. 2g). Similarly, immunofluorescence of tumor tissue also confirmed a higher level of siFXR1-LNA in tumor tissues compared to

**Fig. 2 | LNA-modified FXR1-siRNA exhibited better cellular uptake and improved stability at intracellular level. a** OVCAR8 cells were transfected with siCont-LNA, siFXR1 or siFXR1-LNA were grown as spheroids. Representative images of spheroids were captured from five random fields of three biologically independent experiments with similar results. Scale bars represent 300 μm. **b** Representative images of three biologically independent experiments with similar results show the uptake of Texas red labeled-siFXR1 or siFXR1-LNA in 3D spheroids of GFP-labelled OVCAR8 cells. Scale bars represent 50 μm. **c** Representative images of three biologically independent experiments with similar results of ovarian cancer cells were transfected with Texas red labelled native siFXR1 or siFXR1-LNA for cellular uptake and intracellular distribution of siRNAs were captured at the indicated time points using confocal microscopy. The nucleus was stained with Hoechst, the lysosome vesicles were labeled by Lysotracker Red, and the siRNA was labeled with Texas red. Scale bars represent 50 μm in unzoomed images and 20 μm in zoomed images**. d** Representative images of TUNEL-stained ovarian cancer cells were captured 48 h after siRNA transfection. Green fluorescence indicates TUNEL stained nuclei (top). Scale bars represent 50 μm. Bar graph shows number of apoptotic cells determined by TUNEL assay (bottom) (*n* = 3, represents three biologically independent samples). Significance was determined by unpaired two-tailed Student's *t*-test (****p < 0.0001). **e** Representative agarose gel images show the stability of native or LNA form siRNA either as naked or incorporated in jetPEI nanocomplex following the treatment with or without RNase

I for 30 min. siRNA band intensity was quantified to calculate the percentage of siFXR1 remaining compared to respective control (*n* = 3, represents three biologically independent samples). Significance was determined by unpaired two-tailed Student's *t*-test (**p < 0.01, ****p < 0.0001). **f** Representative PAGE images from three biologically independent experiments with similar results show the levels of indicated siRNAs either as naked or incorporated in jetPEI nanocomplex after incubating in 10% human serum (HS) or mouse serum (MS) at 37 °C for the indicated time points. **g** Representative IVIS images of tumor-bearing mouse (*n* = 3 mice per group) show the biodistribution of Texas Red labelled siFXR1-3 or siFXR1-3-LNA after 1 h of injection; (left panel). Mice were then sacrificed, and tumor tissues from the peritoneal cavity (marked by dashed rectangle) and the indicated organs with tumor metastasis were isolated and imaged for Texas Red fluorescence for biodistribution. Heart and lungs without any metastasis observed were used as a negative control (right panel). **h** Representative confocal microscopy images (top) of tumor tissues were collected from mouse treated with native siFXR1-3 or siFXR1-3-LNA. Cell nuclei were stained for DAPI (blue), and siRNA was labeled with Texas Red (red). Scale bars represent 100 μm. Bar graph showing the average intensity of Texas red in tumor tissue sections (bottom) (*n* = 3). Data represent mean ± SEM of three biological replicates. Significance was determined by unpaired two-tailed Student's *t*-test. (***p < 0.001). Exact *p*-values for (**d**, **h**) are included in Source data file 2.

native siFXR1 (Fig. 2h). Collectively, our findings demonstrate that LNA modification of FXR1 siRNA not only protects FXR1 siRNAs from RNase and nuclease, but also enhances its tumor-specific uptake, suggesting improved therapeutic potential for treating cancer in preclinical models.

## In vivo administration of LNA-modified FXR1 siRNA effectively suppresses ovarian cancer growth and metastasis

In this aim, we delivered LNA-modified siFXR1 or its native form complexed with jetPEI intraperitoneally (IP) to mice bearing ovarian cancer cells. Briefly, OVCAR8 cells were injected IP in female athymic nude mice, and siRNA encapsulated in jetPEI was delivered into the mice twice a week for five weeks after one week of tumor inoculation, as depicted in the schema (Fig. 3a). The control group of mice was treated with LNA modified scrambled siRNA (siCont-LNA) encapsulated in jetPEI as above. Bioluminescence imaging was performed once a week to monitor tumor growth. Notably, mice that were treated with native siFXR1 showed a markable reduction in the overall tumor growth, whereas LNA modified siFXR1 displayed a greater level of reduction in the overall tumor growth compared to the native form of siRNAs (Fig. 3b–d and Supplementary Fig. 4c).

To evaluate the silencing effect of the siRNA complex in vivo, we isolated proteins from the tumor specimen and evaluated the protein levels of FXR1 and its target MYC oncogene. As shown in Fig. 3e, tumors treated with the siFXR1-LNA exhibited robust inhibition of both FXR1 and MYC protein compared with tumors treated with jetPEI encapsulated siCont-LNA. Next, we validated the target specific effect of FXR1 siRNA on the downstream effectors of apoptosis in tumor tissues isolated from mice treated with control, native, or LNA-modified siRNAs by immunohistochemistry (IHC) analysis. IHC of tumor tissues demonstrated that LNA-modified FXR1 siRNA exhibited a markable effect on inhibiting Ki67 proliferation marker levels and a subsequent upregulation in the levels of apoptosis marker cleaved-caspase 3 (Fig. 3f). TUNEL assay also demonstrated extensive TUNEL staining, as an indication of tumor cell death, in the tumor tissues were isolated from mice treated with siFXR1-LNA compared with those treated with native siFXR1 (Supplementary Fig. 4d). Notably, there was no indication of tumor cell death when treated with siCont (Supplementary Fig. 4d).

Next, we validated the tumor inhibitory effects of siFXR1-LNA in a subcutaneous xenograft model using a highly aggressive triple-

negative breast cancer cell line MDA-MB-231 which expresses high levels of FXR1 (Fig. 3g and Supplementary Fig. 4e). Consistently, siFXR1-3-LNA demonstrated a more effective knockdown of FXR1 and its downstream target MYC in MDA-MB-231 cells compared to siFXR1-2-LNA (Supplementary Fig. 4f). We selected an MDA-MB-231 TNBC model due to its genomic similarities with High-Grade Serous Ovarian Carcinoma (HGSOC) including frequent TP53 and BRCA mutations, high genomic instability and the prolific growth and metastatic behavior of the MDA-MB-231 cell line.

Consistent with the response observed in OVCAR8 tumors, systemic retro-orbital injections of siFXR1-LNA also significantly inhibited tumor growth of MDA-MB-231 tumors compared to the unmodified siFXR1 treatment (Figs. 3h–j). Furthermore, histological analysis of lymph nodes used as indicators of early metastasis also revealed no signs of metastasis in the mice that were treated with siFXR1-LNA (Fig. 3k). We further examined whether the delivery of native or LNA modified FXR1 siRNA causes any tissue damage or associated toxicity by performing H&E-staining of liver, lung, heart, and kidney sections. Notably, neither form of FXR1 siRNA induced any observable histological changes compared with the respective control, indicating no detectable tissue damage associated with cellular toxicity (Supplementary Fig. 4g). We also assessed key enzymes and proteins for liver and kidney functions including Aspartate Aminotransferase (AST); Alanine Aminotransferase (ALT); Alkaline Phosphatase (ALP); Total Bilirubin (TBIL); Total Protein (TP) and found that delivery of both native and LNA modified FXR1 siRNA caused no significant changes in the levels of these markers (Table S2).

Taken together, results from our in vitro and in vivo models, along with toxicity assessments, demonstrate that LNA-modified FXR1 siRNA effectively inhibits tumor growth and metastasis by silencing its target FXR1 with minimal or no adverse effects on tissue integrity or organ functions.

## Single-cell transcriptome profiling demonstrates marked effects of LNA-modified FXR1 siRNA on tumor growth inhibition and anti-tumor immune responses

One of the major challenges of translating RNAi-based therapies into clinical practice is the risk of immune cell toxicities and unintended effects on stromal components within the tumor microenvironment (TME). Therefore, we performed scRNA-seq method to assess whether there were any adverse effects on transcriptomic changes induced by

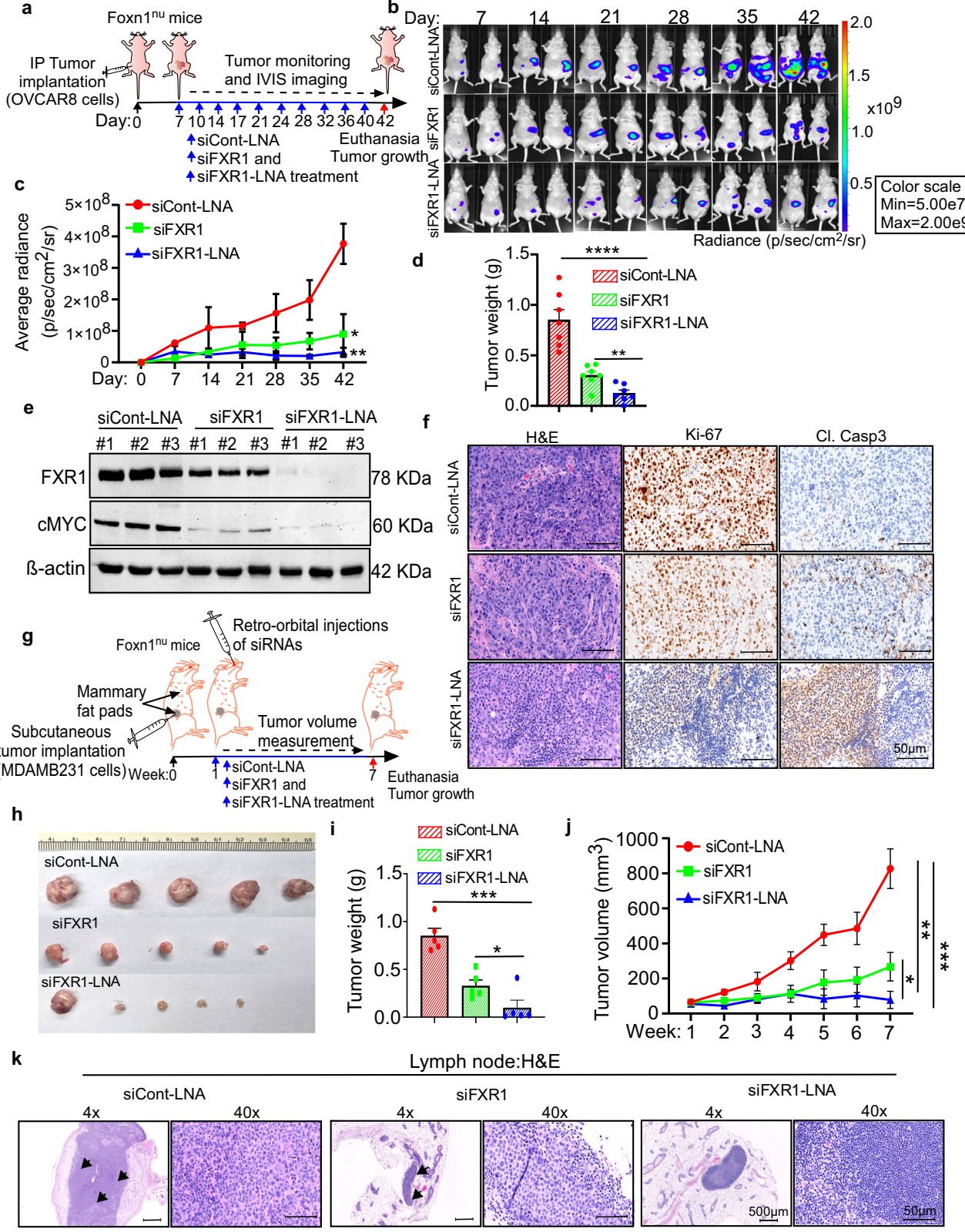

RNAi treatment in both tumor cells and the cells in TME comprehensively. For this goal, we used both native and LNA-modified forms of FXR1-targeting siRNAs (seq2 or seq3) to knockdown FXR1 in mouse ovarian cancer cell lines, where seq2 showed 92% sequence complementarity and seq3 showed 96% complementarity with mouse FXR1 mRNA (Supplementary Fig. 5a). Consistent with the target knockdown effects in human cell lines and high level sequence complementarity,

seq3 siRNA showed superior level of FXR1 inhibition in two mouse ovarian cancer cell lines Br-Luc (genotype: p53$^{-/-}$; brca1$^{-/-}$; myc; Akt-myr) and C11 (genotype: p53$^{-/-}$; myc; Kras$^{G12D}$)[37]. Markedly, the LNA-modified versions exhibited more robust inhibition of FXR1 levels compared to unmodified versions (Supplementary Fig. 5b).

The Br-Luc cell line was selected because of its mutational similarity to human HGSOC for establishing an immunocompetent

**Fig. 3 | In vivo delivery of LNA-modified FXR1 siRNAs exhibited superior level of inhibition of the growth and metastasis of ovarian and breast cancer cells.** **a** Schema shows the schedule of injections of jetPEI-incorporated siCont-LNA, native siRNA or siFXR1-LNA injected (twice/week) intraperitoneally (IP) in OVCAR8 tumor-bearing female athymic nude mice ($n = 7$ mice per group). On day 42, all mice were euthanized, and tumors were collected for further analysis. **b** Mice from **a** were imaged using an IVIS imager and representative images of two mice per group were presented at the indicated time point. **c** Bioluminescent signals from **b** were quantitated at the indicated time points and presented. Significance was determined by unpaired two-tailed Student's $t$-test on the last time point (**$p < 0.01$, ***$p < 0.001$). **d** Primary and disseminated tumors were collected from **b**, and total tumor weight was recorded (right). Significance was determined by unpaired two-tailed Student's $t$-test (**$p < 0.01$, ****$p < 0.0001$). **e** Western blot analysis of indicated proteins in the lysates from representative tumor tissues collected from each treatment group ($n = 3$ biologically independent samples for each). β-actin, loading control. Uncropped blots are provided in Source Data file 1. **f** Representative images of H&E stained or Ki67 and cleaved Caspase3 (Cl. Casp3) immunostained tumor tissue sections from three biological independent animals from indicated groups. Scale bars represent 50 μm. **g** Schema shows the schedule of injections of jetPEI-incorporated siCont-LNA, native siRNA, or siFXR1-LNA injection (10 μg/dose; twice/week) IP for 6 weeks in MDA-MB-231 tumor-bearing female athymic nude mice. In week 7, all mice were euthanized, and tumors were collected and photographed. **h** Tumors from each group were isolated and photographed at the end point; ($n = 5$ mice per group). **i** Bar graph represents tumor mass in grams quantitated at the end point. Significance was determined by unpaired two-tailed Student's $t$-test (*$p < 0.05$, ***$p < 0.001$). **j** Tumor volume in mm³, was measured at the indicated time point for six weeks after tumor cells inoculation ($n = 5$ mice per group). Significance was determined by unpaired two-tailed Student's $t$-test on the last time point (*$p < 0.05$, **$p < 0.01$, ****$p < 0.0001$). **k** Representative H&E staining images captured from five random fields of lymph nodes isolated from three mice from each treatment group and presented. Metastatic regions are marked with arrows. Scale bars represent 500 μm and 50 μm respectively at 4x and 40x magnification. Mice drawings in **a**, **g** were created using Canvas X Pro. Exact $p$-values for (**c**, **d**, **i**, **j**) are included in Source data file 2.

syngeneic model of ovarian cancer through IP injection of Br-Luc cells into immunocompetent FVB mice[38]. Mice were treated with siCont-LNA or siFXR1-LNA encapsulated in jetPEI nanoparticles (10 μg/mouse) twice a week for five weeks (Supplementary Fig. 5c). In conjunction with our human xenograft model, we observed a significant reduction in both tumor weight and ascites formation when the mice were treated with jetPEI encapsulated siFXR1-LNA (Supplementary Fig. 5d–f). We also observed that siFXR1-LNA treatment significantly improved the survival of Br-Luc ovarian cancer bearing mice compared with the control siRNA-LNA treated group (Supplementary Fig. 5g).

Ascites samples from tumor-bearing mice treated with the siCont-LNA and siFXR1-LNA were used for scRNA-seq, followed by clustering and CellChat analysis to delineate the role of various cell populations within the TME on tumor progression (Fig. 4a). It has been reported by other scRNA-seq experts that ascites represents a clinically valuable source of transcriptional information of both tumor cells and tumor-associated immune and stromal cells[29,30]. These cells can be isolated through minimal invasive procedures, and cell clusters within ascites can be more readily and easily dissociated into single cells compared to solid tumors. Moreover, ascites often reflects advanced metastatic disease and immunosuppressive signaling, providing critical insights into mechanisms of how tumors escape from immune surveillance or therapies[39]. In brief, 16851 high-quality cells (~11621 cells from siCont-LNA and ~5230 cells from siFXR1-LNA group) were selected for generating single-cell transcriptomic profiles (Supplementary Fig. 5h).

The cell clusters were then visualized using Uniform Manifold Approximation and Projection (UMAP) for the integrated duplicates performed at a resolution of 0.1 annotation level according to sample type with differentially expressed genes (DEGs) (Supplementary Fig. 5i). Integrative analysis identified 11 clusters of cells in the murine TME components in the ascites (Fig. 4b and Supplementary Fig. 5j). Among the cell types, macrophages (cluster 0), epithelial cells (clusters 1, 4), T cells (cluster 3), NK cells (cluster 7), and DCs (cluster 8) showed high level differences between clusters among siCont-LNA and siFXR1-LNA groups (Fig. 4b and Supplementary Fig. 5j). Overall, the composition of cell clusters was variably influenced by siFXR1-LNA treatment compared to the siCont-LNA group (Fig. 4c).

Interestingly, we observed a notable reduction in epithelial cells and a substantial increase in DCs, NK, and T cells in the siFXR1-LNA treated group (Fig. 4c), indicating enhanced immune cell infiltration and a decrease in overall tumor growth potentially due to the direct inhibition of FXR1 in tumor cells as well as the effect through improved immune cell infiltration. While we identified distinct markers among the cells belonging to main clusters in Fig. 4b–d, further analysis identified considerable heterogeneity among the cells present in each cluster. For example, cluster 0 represents macrophage populations

since many of the cells in this cluster express genes like *Cd68, Arg1, Adgre1, Apoe,* and *Lyz2* (Fig. 4c, d and Supplementary Fig. 5j), but a considerable level of heterogeneity was observed in this population. Likewise, clusters 1 and 4 were annotated as epithelial cells (tumor cells) based on the high expression of *Krt19, Krt18, Krt8, Epcam* and *Muc16* in those populations (Fig. 4c, d and Supplementary Fig. 5j). Cells in cluster 3 were identified as T cell based on *Lef1, Cd3e, Cd3g, Cd3d, Cd4* and *Cd8a* expression and cluster 5 was determined as B cell based on *Cd19, Bank1, Ebf1, Igkc* and *Ighm* expression (Fig. 4c, d and Supplementary Fig. 5j). Cluster 7 showed expression of canonical markers of NK cells expressing *Gzmb, Ccl5, Nkg7, Ncr1* and *Ilr2b* while cluster 8 was classified as DCs due to the presence of *Siglech, Cd74, Grm8, Bcl11a, Itgae,* and *Clec9a* (Fig. 4c, d and Supplementary Fig. 5j). Clusters 9 and 10 were relatively small populations and marked as mast cells and fibroblast respectively (Fig. 4c, d and Supplementary Fig. 5j).

Because we identified a considerable decrease in the numbers of epithelial tumor cells and an increase in immune cells (clusters 0, 1, 3, 4, 7, and 8, Fig. 4b–d), we performed CellChat analysis to determine the potential ligand-receptor interactions among cell types[40]. Notably, the interactions between tumor cells to macrophages and other immune cells were very high in the control group, which significantly reduced upon siFXR1-LNA treatment (Fig. 4e and Supplementary Fig. 5k). Furthermore, communications of T and NK cells with tumor cells were greatly enhanced upon siFXR1-LNA treatment compared to the control, suggesting an anti-tumor response mechanism. In addition to T and NK cells, an increase in the interaction between DCs, B cells, and tumor cells was observed in the siFXR1-LNA group (Fig. 4e and Supplementary Fig. 5k). This data prompted us to determine the level of cellular interaction by quantitating the magnitude of interaction between ligands and receptors expressed by each cell population using CellChat analysis.

In the CellChat, all the ligand-receptor interactions were grouped into ECM-receptor interaction, secreted signaling, and cell-cell contact. We found that ECM-receptor signaling is downregulated and secreted signaling and cell-cell contact signaling are up-regulated in the siFXR1-LNA treated group compared to the siCont-LNA group (Supplementary Fig. 6a). Mostly the ECM signaling regulated through collagen, laminins, and fibronectin were predominant in the control group, where they potentially act as an immunosuppressive signaling through paracrine mechanism or as an oncogenic signaling in tumor cells through autocrine mechanism. In contrast, such signaling was not present in the siFXR1-LNA treated group. Specifically, collagens (encoded by *Col4a1, Col4a2, Col4a5,* and *Col4a6*), laminins (*Lama3* and *Lamc1*), and fibronectin (*Fn1*) were expressed by epithelial cells and the receptor partner such as various integrins, *Cd44,* and *Sdc4* were observed in tumor cells, and immune cells in the control group

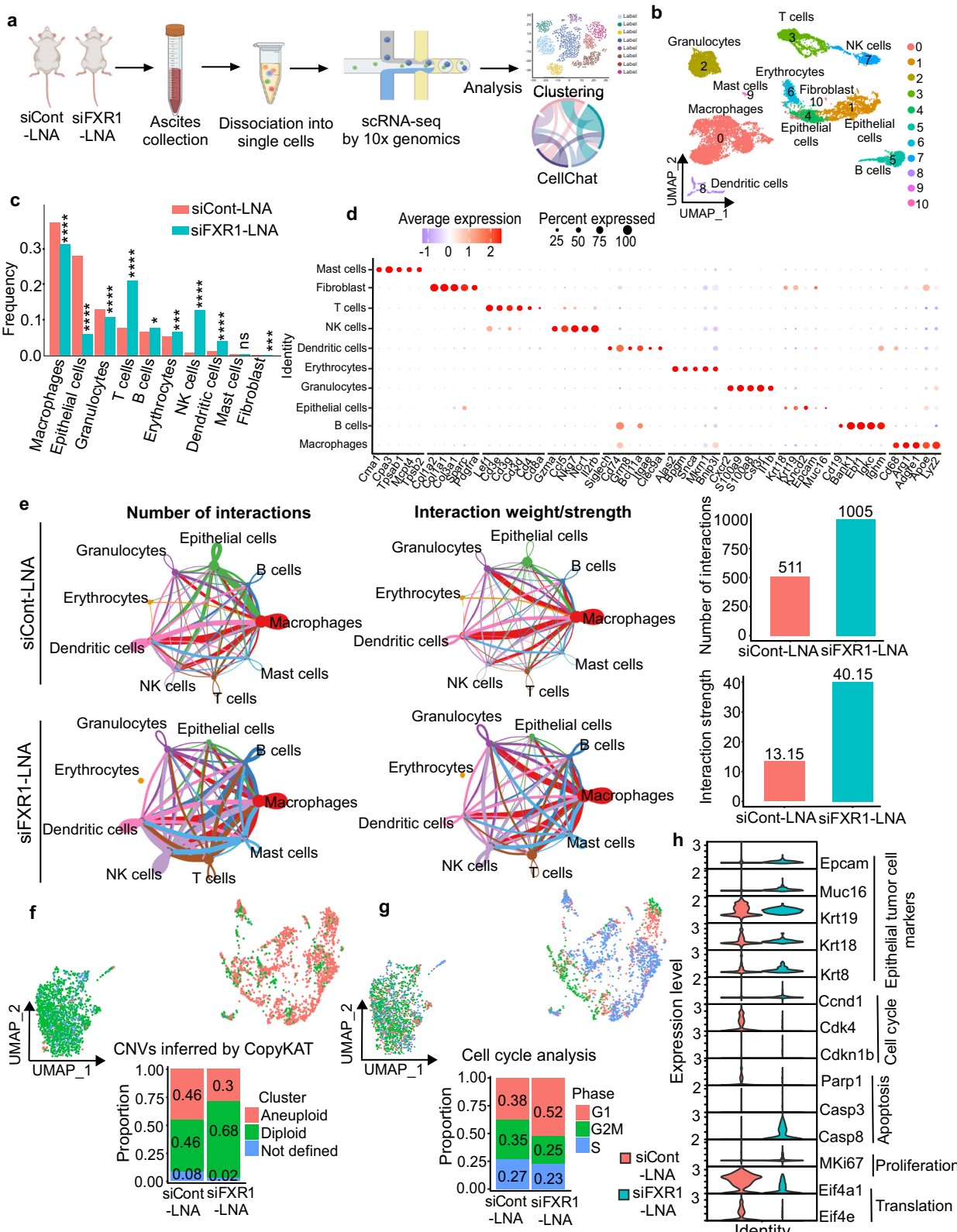

(Supplementary Fig. 6b). Other ECM-related outgoing signaling from macrophages, involving Fn1 and its interactions with integrins, *Cd44*, and *Sdc4* receptors, were observed in both siFXR1-LNA and siCont-LNA treated groups (Supplementary Fig. 6b).

We also observed a gain in cell-to-cell contact signaling through MHC-I/II signaling components such as *H2-t23, H2-aa, H2-ab1, H2-dma, H2-eb1, H2-oa,* and *H2-ob* among immune cells with notable magnitude

from macrophages, DCs, B cells in partnership with T and NK cells mainly in the siFXR1-LNA treated group. In contrast, we did not observe such interactions in the siCont-LNA group (Supplementary Fig. 6c). This data further confirms that macrophages, DCs, and B cells are actively engaged as antigen-presenting cells (APCs) and deliver MHC-II molecules to T and NK cells for anti-tumor responses in the siFXR1-LNA treated group (Supplementary Fig. 6c). Similarly, the

**Fig. 4 | scRNA-seq analysis demonstrated that FXR1 siRNA treatment enhances the anti-cancer immune response and reduces the tumor cell population within epithelial cell clusters in vivo. a** Schema shows the overall study design of sample collection adapted for 10x Genomics scRNA-seq analysis using ascites samples collected from jetPEI encapsulated siCont-LNA or siFXR1-LNA treated mice. Ascites from 3 mice were pooled in each group. Figure was partially created using BioRender image illustration software. **b** UMAP plot shows 11 clusters of cell populations identified from siCont-LNA or siFXR1-LNA treated groups. Each dot corresponds to a single cell, colored by major cell types in clusters. **c** A comparative quantitative bar graph indicates the frequency of each cell type in both groups. *p*-values were determined using the two-sided Wilcoxon Rank Sum test between respective groups (\**p* < 0.05, \*\*\**p* < 0.001, \*\*\*\**p* < 0.0001, ns non-significant). **d** Dot plot shows signature genes among the 11 clusters. The size of the dots indicates percentage of cells expressing the gene, while the color of the dots indicates the average gene expression level. **e** Circle plot shows the number and strength of interactions among cell populations generated by CellChat in siCont-LNA or siFXR1-LNA treated sample groups. Edge color represents signaling direction, and edge thickness indicates communication strength (left). Bar graph shows the comparison of the number and strength of interaction among cell populations in each group (right). **f** Integrated UMAP plot and proportion bar plot shows the chromosomal copy number variation (CNV) status in epithelial cells from the siCont-LNA or siFXR1-LNA groups. *p* values were determined using the two-sided Wilcoxon Rank Sum test between respective groups. **g** Integrated UMAP plot and proportion bar plot shows the distribution of cell cycle phases in epithelial cells from the siCont-LNA and siFXR1-LNA groups. *p* values were calculated by two-sided Wilcoxon Rank Sum test. **h**, Violin plot represents the expression for indicated genes in siCont-LNA and siFXR1-LNA, and *p*-values were calculated by two-sided Wilcoxon Rank Sum test. Exact *p*-values for (**c**, **f**, **g** and **h**) are included in Source data file 2.

control group showed low expression of ligands like *Cd40lg, Cd80*, and *Cd86* with their respective receptors *Cd40* and *Cd28*, which are important for activating T and NK cells (Supplementary Fig. 6c). Moreover, APCs like macrophages, B cells, and DCs in the siFXR1-LNA group exhibited robust *Cd80-Cd28* and *Cd86-Cd28* interaction, suggesting that T cells and NK cells are the primary recipients of signals from APCs, which again confirmed the existence of strong anti-tumor effects upon siFXR1-LNA treatment (Supplementary Fig. 6c). *Cd40-Cd40lg* interaction was also observed in high level in the siFXR1-LNA group (Supplementary Fig. 6c), which is known for T cell proliferation, survival, and differentiation, as well as for enhancing the antibody production capabilities of B cells as reported before[41].

*Il-18* interactions with Il-18r1 and *Il-8* receptor accessory protein (Il-18rap) through macrophages to NK and T cells act as the recipients were also observed to be high in the siFXR1-LNA group suggest Il-18 mediated activation of NK and T cells (Supplementary Fig. 6d). *Tgf-β*-associated cellular interactions, such as Tgfb1 signaling through the *Tgfbr1* and *Tgfbr2* receptors, as well as Tgfb1 signaling through the *Acvr1* and *Tgfbr2* receptors as immune cell activation, were also observed high in the siFXR1-LNA treated group (Supplementary Fig. 6d).

**Targeted silencing of FXR1 in vivo efficiently suppresses oncogenic signaling mechanisms and inhibits tumor growth**

scRNA-seq analysis identified that the control siRNA treated group had a large number of epithelial cells that includes tumor cells, whereas siFXR1-LNA treatment reduced the numbers of epithelial cells (Fig. 4c). Considering that the epithelial cells could consist of both tumor cells and nonmalignant epithelial cells, we used the CopyKAT algorithm[42] to identify tumor cells based on aneuploid status, determined by copy number variation (CNV). As expected, the proportion of tumor cells (aneuploid) were very high over normal epithelial cells (diploid) in the control group, whereas the aneuploid cell numbers were markedly diminished without affecting diploid cells when treated with siFXR1-LNA (Fig. 4f and Supplementary Fig. 7a). Consistently, flow cytometry of ascites and immunofluorescence analysis of tumor tissues demonstrated a reduction in EpCAM+ cells accompanied by a markable increase in Cd45+ immune cells in siFXR1-LNA treated samples compared to the siCont-LNA group (Supplementary Fig. 7c, d).

Additionally, transcriptome profile of cell cycle phase also showed a decrease in S phase cells and an increase in G1 phase genes in siFXR1-LNA treated group compared to those treated with control siRNA (Fig. 4g and Supplementary Fig. 7b). We also observed down-regulation of tumor cell markers, including *Epcam, Muc16, Krt19, Krt18*, and *Krt8*, following siFXR1-LNA treatment (Fig. 4h). Furthermore, a significant downregulation of genes associated with G1-phase such as *Ccnd1* and *Cdk4*, along with a notable upregulation of the cell cycle arrest gene *Cdkn1b* (*p27Kip1*) were observed in the cells belonging to the siFXR1-LNA treated group (Fig. 4h). We also observed

a significant decrease in the expression of the apoptosis-related gene *Parp1*, along with an increase in the expression of *Casp3* and *Casp8* upon FXR1 silencing (Fig. 4h). A reduction in the expression of the proliferation marker gene *MKi67* (Fig. 4h) and genes associated with DNA replication such as *Top2a* and *Mcm6* were also observed upon siFXR1-LNA treatment (Supplementary Fig. 7e). Eukaryotic translation-related genes *Eif4g1* and *Eif4e*, which are the direct targets of FXR1, were also markedly reduced after siFXR1-LNA treatment (Fig. 4h). Most importantly, *Fxr1* expression was significantly downregulated in siFXR1-LNA treated epithelial cells compared to those treated with control siRNA (Supplementary Fig. 7f), again confirmed the specificity of siFXR1-LNA on silencing FXR1in tumor cells.

Ingenuity pathway analysis (IPA) of epithelial cluster genes (Supplementary Data 1) further revealed enrichment of gap junction signaling, cell cycle control of chromosomal replication, cell cycle regulation etc., in the siFXR1-LNA group compared to control (Supplementary Fig. 7g). Consistent with our previous work reported the role of FXR1 in oncogenic translation mechanism[9], we observed FXR1 silencing significantly suppressed the expression of genes involved in translation initiation, elongation and termination in the tumor cells compared to the controls (Supplementary Fig. 7g). Pathways related to mTOR, Keap-Nfe2l2, and PPAR family signaling pathways, which are crucial for oncogenic metabolism[43–45] were also downregulated upon treatment with siFXR1-LNA (Supplementary Fig. 7g).

**In vivo FXR1 silencing enhances anti-tumor response via macrophage polarization and dendritic cell infiltration**

Tumor associated macrophages (TAMs) are known for their effects in cancer progression and metastasis[46–50]. TAMs are also known for inducing immune suppressive effects on T cells and NK cells by modulating their functions either by secreting anti-inflammatory cytokines, chemokines, or by modulating the functions of immune checkpoint proteins[51–54]. Notably, our primary analysis identified macrophages as the dominant cell population in cluster 0, comprising ~35% of the total cell population (Fig. 4b, c). UMAP clustering of this population at 0.1 resolution generated about 3 distinct clusters as M1-like, M2-like, and M1/M2/monocyte mixed lineage population in the siCont-LNA and siFXR1-LNA groups (Fig. 5a, b and Supplementary Fig. 8a). In brief, cluster 0 were designated as M2-like TAMs with high expression of *Arg1, Cd163, Mrc1*, and *Adgre1* and cluster 1 was identified as M1-like macrophages expressing anti-tumor factors including *Ccl2, Ccl3, Ccl7, Cxcl9, Cxcl10*, and *Cxcl16* (Fig. 5a, b). The remaining cells in cluster 3 was identified as mixed lineage of M1/M2/monocyte with the presence of monocyte markers *Lyz1, Ly6c2*, and *Cd14* along with M1 and M2 markers (Fig. 5a, b).

Next, we examined the expression of *Ccl2* and its receptor *Ccr2*, as well as *Ccl7, Cxcl10, Ccl5*, and its receptor *Ccr5* in each macrophage subtype. M1-like macrophages (cluster 1) exhibited significantly higher

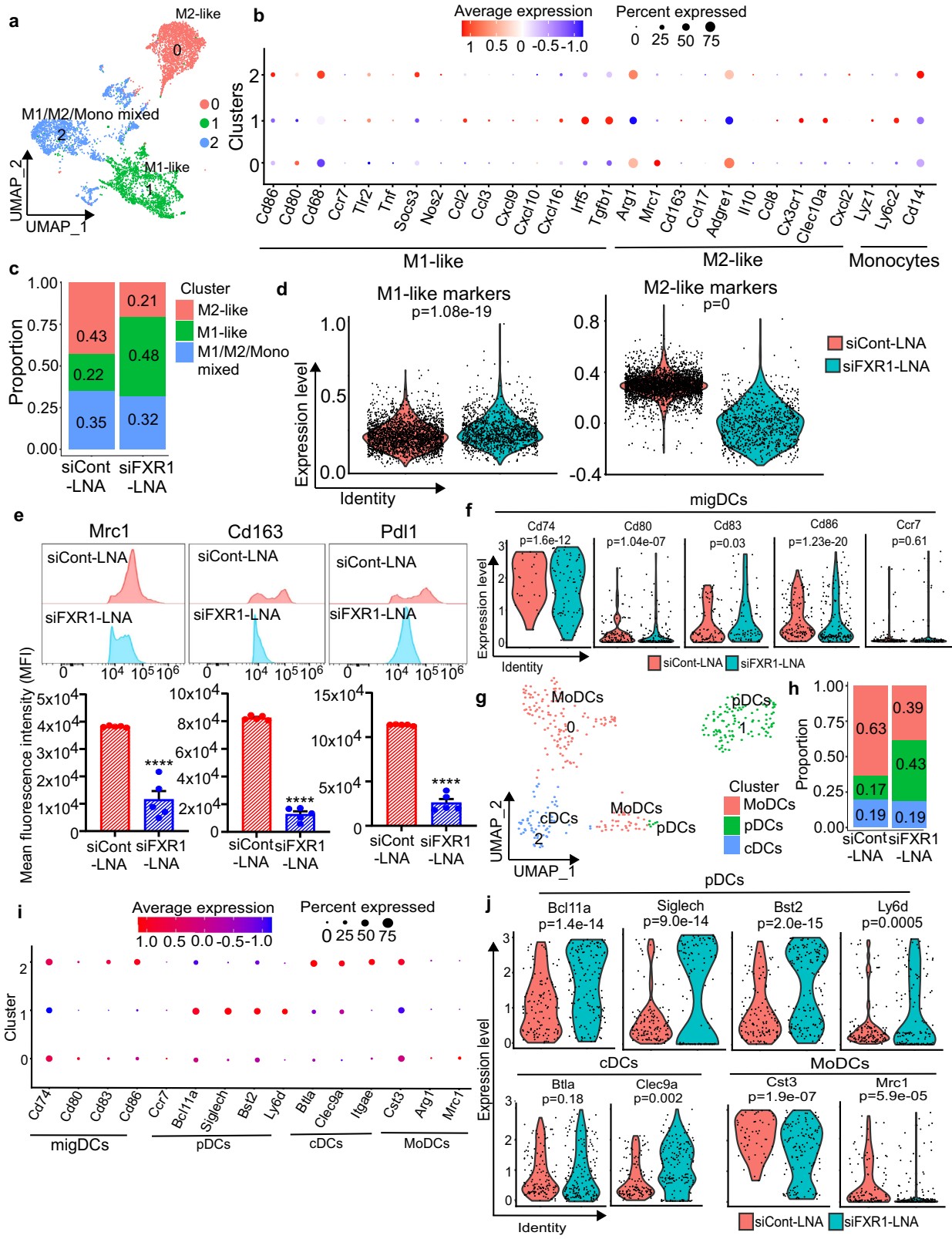

expression of these genes compared to M2-like macrophages (cluster 0), which facilitates the recruitment and activation of T cells (Supplementary Fig. 8b). While tumor promoting M2-like macrophage populations were enriched over M1-like macrophages in the control group, siFXR1-LNA treatment resulted in enrichment of M1-like macrophages with less abundance of M2-like macrophages (Fig. 5c and Supplementary Fig. 8c). To further characterize the macrophage population

in detail, we computed M1-like and M2-like signatures score using gene signature scores based on gene sets derived from CIBERSORT[55]. These scores were then integrated to generate module scores using the AddModuleScore function. As anticipated, the M1-like macrophage signature was significantly higher, and the tumor promoting M2 macrophage signature scores were very low in the siFXR1-LNA treated group compared to the controls (Fig. 5d). To further validate the

**Fig. 5 | FXR1 siRNA improved macrophage and dendritic cells (DCs) population with anti-tumor characteristics in vivo. a** UMAP plot shows subclusters of macrophages cell populations identified in the samples that were collected from siCont-LNA or siFXR1-LNA treated groups. **b** Dot plot shows the marker genes macrophage subtypes across the clusters. The dot size denotes the percentage of cells expressing the marker gene in the respective cell population, and the color scale denotes the level of normalized RNA expression. **c** Stacked bar graph illustrates the proportion of distribution for macrophage subtypes across three subclusters in siCont-LNA or siFXR1-LNA treated groups, *p*-values were calculated by two-sided Wilcoxon Rank Sum test. **d** Violin plot shows M1 like macrophage signature score and M2 like macrophage signature score in macrophage populations in siCont-LNA or siFXR1-LNA treated groups. *p*-values were calculated by two-sided Wilcoxon rank sum test. **e** Representative histograms (upper panel) show the levels of indicated marker proteins in TAM subpopulations in siCont-LNA and siFXR1-LNA

treated groups determined by flow cytometric analysis. Mean fluorescence intensity (MFI) of indicated markers were quantitated and presented below (*n* = 5, represents five biologically independent samples). Error bars indicate mean ± SEM. Significance was determined by unpaired two-tailed Student's *t*-test (****$p$ < 0.0001). **f** Violin plot shows expression for indicated genes in siCont and siFXR1-LNA treated groups. *p*-values were calculated by two-sided Wilcoxon rank Sum test. **g** UMAP plot shows subclusters of DCs in siCont-LNA or siFXR1-LNA treated groups. **h** Stacked bar graph illustrates the proportion of DC subtypes across three subclusters in siCont-LNA or siFXR1-LNA treated groups. *p*-values were calculated by two-sided Wilcoxon Rank Sum test. **i** Dot plot shows proportions of each DC subtype across three subclusters. **j** Violin plot shows expression for indicated genes in siCont-LNA or siFXR1-LNA treated groups, *p*-values were calculated by two-sided Wilcoxon Rank Sum test. Exact *p*-values for figures (**c**, **e**, **h**) are included in Source data file 1.

transcriptomic data identified by scRNA-seq, we assessed the protein levels of M2 macrophage markers using flow cytometry and immunofluorescence. Consistent with the scRNA-seq data, flow cytometry revealed enrichment of macrophages with M2-like markers *Mrc1*, *Cd163* and *Pdl1* in the control group, while siFXR1-LNA treatment significantly reduced the expression of these markers significantly (Fig. 5e and Supplementary Fig. 8d). We also observed a significant decrease in *Pdl1*[+] TAMs in the siFXR1-LNA treated group, which again confirms the anti-tumor effects of siFXR1-LNA (Fig. 5e and Supplementary Fig. 8d). Immunofluorescence of tumor tissues further confirmed a decrease in the infiltration of M2 macrophages as evidenced by low expression of *Arg1* and *Mrc1* in the tumor samples isolated from mice were treated with siFXR1-LNA compared to the controls (Supplementary Fig. 8e).

Next, we analyzed the dendritic cells (DCs), which are known as the sentinels of the immune system and could be modulated for immunotherapeutic treatment strategies[56]. Thus, we analyzed the population of DCs for their antigen-presenting capabilities, such as recognizing, processing, and presenting "threat signals" obtained upon siFXR1-LNA treatment. Our analysis identified distinct expression profiles of DCs isolated from mice treated with control siRNA and siFXR1 LNA (Supplementary Fig. 8f). Notably, DCs expressing high levels of genes associated with DC maturation and migration, namely *Cd74, Cd80, Ccr7, Cd83*, and *Cd86* were observed in the group treated with siFXR1-LNA (Fig. 5f). Because these cells express a specific gene signature for DC activation and maturation, we performed a more detailed subcluster analysis and found three distinct subclusters of DCs in the tumor (Fig. 5g). Cluster 0 was identified as the monocyte derived DCs (MoDCs) expressing *Cst3, Mrc1* and *Arg1* genes, cluster 1 as plasmacytoid DCs (pDC) expressing *Bcl11a, Siglech, Bst2* and *Ly6d* and cluster 2 as type1 conventional DCs (cDC1) expressing *Btla* and *Clec9a* (Fig. 5g–j). We also observed that the ratio of MoDCs, a subset of DCs that differentiate in response to inflammatory stimuli and are recruited to inflammatory sites such as the TME, was significantly reduced in siFXR1-LNA treated sample groups (Fig. 5h). Importantly, the proportion of pDCs, which promote anti-tumor immunity via type I interferons (IFN-I)[57,58], and facilitates cDC1 maturation to enhance Cd8[+] T cell and NK cell cytotoxicity[59], was significantly increased in the tumor samples from mice treated with siFXR1-LNA (Fig. 5h). However, no significant changes were observed in the cDC1 population (Fig. 5h).

Next, we performed CellChat analysis to investigate the interactions between pDC and NK cells, as well as Cd8[+] and Cd4[+] cytotoxic T cells, to assess their impact within TME. In the siFXR1-LNA group, pDC ligands (*Klrb1a, Klrb1b, Klrb1c, Klrb1f*) were found to interact with Clec family receptors on NK cells (Supplementary Fig. 8g). These interactions are expected to enhance NK cell activation and functionality, promoting the recognition and destruction of tumor cells[60]. Additionally, pDCs which are known for APC functions by providing signals to Cd8[+] and Cd4[+] cytotoxic T cells through interactions with

MHC class I and II receptors were predominant in the siFXR1-LNA group (Supplementary Fig. 8g). Overall, our analysis demonstrates that these pDCs-mediated interactions, which are important for bridging innate and adaptive immunity, facilitate T cell priming and activation in the presence of tumor antigens enhanced upon siFXR1-LNA treatment.

### In vivo silencing of FXR1 potentiates anti-tumor immunity by improving cytotoxic T and NK cells infiltration

Our primary analysis showed an increase in T and NK cell populations in samples treated with siFXR1-LNA compared to those treated with control siRNA (Fig. 4c). Unsupervised clustering based on expression of *Cd3d, Cd3e*, and *Cd3g* for tumor-infiltrating lymphocytes (TILs) and *Nkg7* for NK cells identified six distinct T cell clusters and three NK cell clusters in our sample sets (Fig. 6a and Supplementary Fig. 9a). Specifically, our T-cell subtyping identified subclusters 0 and 1 as naïve Cd4[+] and Cd8[+] T cells, characterized by expression of *Lef1, Tcf, Sel and Ccr7*, while subclusters 2 and 3 were identified as cytotoxic Cd4[+] and Cd8[+] T cells marked by *Nkg7* and *Gzmb*. Subcluster 4 included a small fraction of Cd4[+] exhausted T cells/Tregs characterized by *Tigit/Foxp3* expression, and subcluster 5 was identified as effector memory Cd4[+]/Cd8[+] T cells, marked by Isg15 expression (Fig. 6b). Quantification of all cell populations in both groups detected abundance of Cd4[+] T cells over Cd8[+] T and other T cell populations following FXR1 siRNA treatment (Fig. 6c). Most strikingly, we observed a shift in the overall population of naïve Cd4[+] T cells and Cd8[+] T cells along with the cytotoxic form of Cd4[+] T cells and Cd8[+] T cells. In brief, we found a decrease in naïve Cd4[+] T cells and Cd8[+] T cells. In contrast, we found an increase in the enrichment of Cd8[+] T cells and Cd4[+] T cells with cytotoxic function indicated by the marker genes *Nkg7, Prf1*, and *Gzmb* upon FXR1 siRNA treatment (Fig. 6d and Supplementary Fig. 9b).

We also identified a small cluster of Cd4[+] T cells that express markers associated with exhausted T cells and Tregs in our T cell clustering (cluster 4; Fig. 6a, b). However, we did not observe any significant differences in the proportions of Cd4[+] Treg cells between the siFXR1-LNA and control groups (Fig. 6d). In complement to scRNA-seq analysis, our IHC analysis of tumor tissue samples also demonstrated high levels of T cell infiltration in the tumor tissues upon FXR1 silencing compared to controls (Fig. 6e). Additionally, flow cytometry analysis of the ascites samples collected from the mice treated with control or FXR1 siRNA also exhibited an increase in Cd4[+] and Cd8[+] T cells upon siFXR1-LNA treatment compared to the siCont-LNA group (Fig. 6f and Supplementary Fig. 9c). Overall, our findings demonstrate a substantial increase in total T cells, particularly an increase in the cytotoxic forms of Cd4[+], Cd8[+] T cells in the tumor bearing mice treated with siFXR1-LNA.

Next, we performed Cell Chat analysis[40] to determine cell-cell communication between cytotoxic Cd8[+] and Cd4[+] T cells populations with macrophages and tumor cells (epithelial cells), which identified

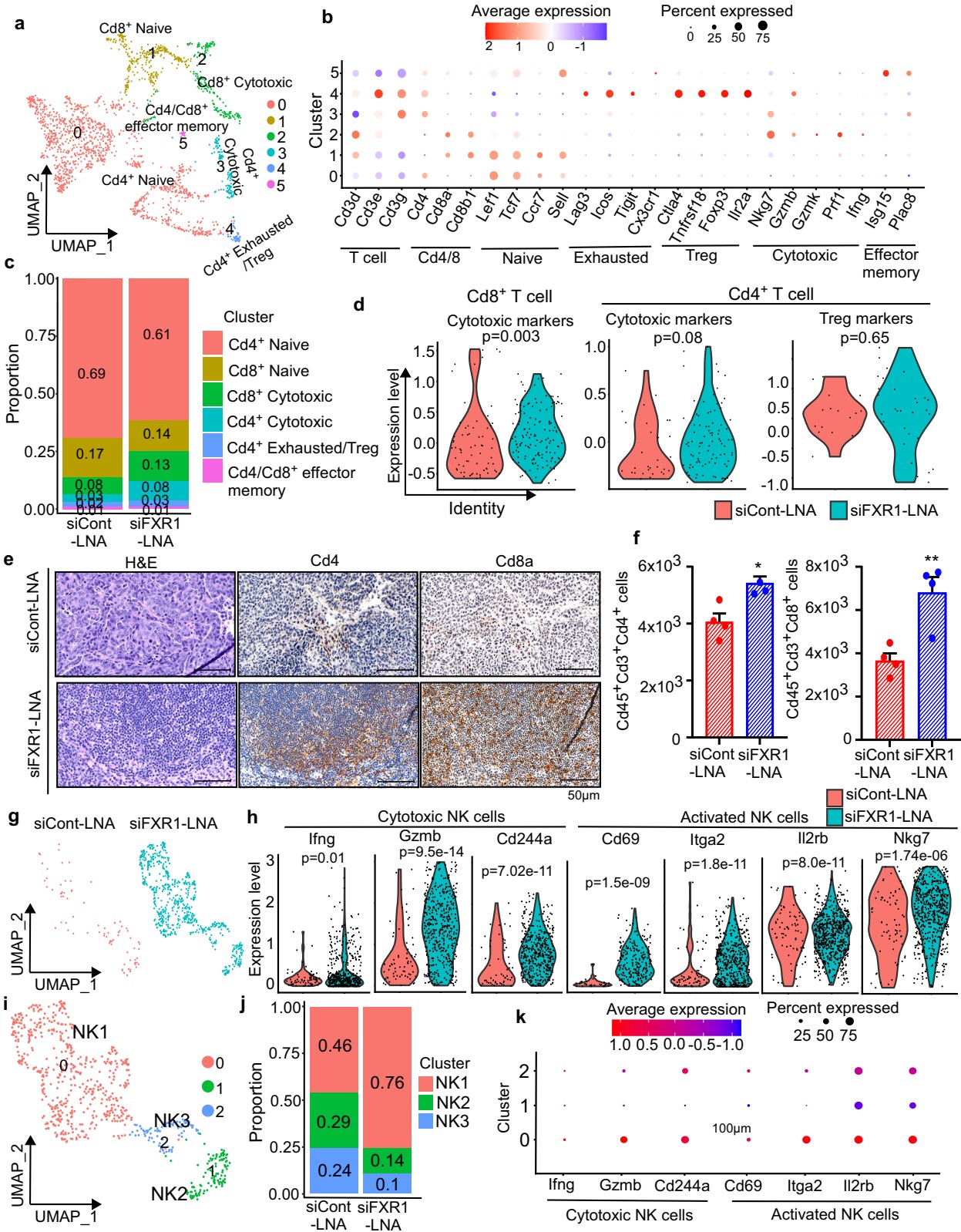

numerous interactions between macrophages, tumor cells and T cells (Supplementary Fig. 9d). Among them, the interaction of *Ccl5* ligand with its receptor *Ccr1* and *Ccr5* between Cd8⁺ cytotoxic and Cd4⁺ cytotoxic T cell subtypes, M1- and M2-like macrophages and tumor cells were observed high as a mechanism of T cell mediated cell death in the siFXR1-LNA treated group compared to the control (Supplementary Fig. 9d). We also found that the interaction of adhesion

molecules such as *Icam1* with *Itgal*, *Itgab2*, and *Itgam*, along with *Thy1's* interaction with *Adgre5*, *Itgam*, and *Itgb2* between macrophages and tumor cells with T cells were observed only in the siFXR1-LNA group as a potential mechanism of T cell activation and recruitment of macrophages into tumor sites for the elimination of tumor cells (Supplementary Fig. 9d). We also observed prominent interactions between *Cd40lg* with *Itga5*, *Itgam*, *Itgb2* and *Cd40*, which are critical for T cell

**Fig. 6 | FXR1 siRNA improved helper and cytotoxic T cells and cytotoxic NK cells within TME. a** UMAP plot shows subclusters of T cells in siCont-LNA or siFXR1-LNA treated groups. **b** Dot plot shows the signature gene expression across the T cell subtypes. The dot size denotes the percentage of cells expressing the marker gene in the respective cell population, and the color scale denotes the level of normalized RNA expression. **c** Stacked bar graph illustrates the proportion of T cell subtypes in siCont-LNA or siFXR1-LNA treated groups. *p*-values were calculated by two-sided Wilcoxon Rank Sum test. **d** Violin plot shows average expression of cytotoxic markers in Cd8⁺ and Cd4⁺ T cells and Treg markers in Cd4⁺ T cells from siCont-LNA and siFXR1-LNA groups, p-values were calculated by two-sided Wilcoxon rank sum test. **e** Representative images captured from five random fields of H&E of tissue sections prepared from each group immunostained for Cd4 and Cd8 proteins (*n* = 3, represents tumor samples used from each group). Scale bars

represent 50 µm. **f** Quantitative bar graph shows the levels of Cd4⁺ and Cd8⁺ T cell infiltration determined by flow cytometry analysis in the tumor ascites collected from siCont-LNA or siFXR1-LNA treated group (*n* = 4, represents ascites samples used from each group). Error bars indicate mean ± SEM. Significance was determined by unpaired two-tailed Student's *t*-test (*$p < 0.05$, **$p < 0.01$) **g** UMAP plot shows abundance of NK cells in the siCont-LNA or siFXR1-LNA treated groups. **h** Violin plot shows the expression of indicated genes in siCont-LNA or siFXR1-LNA treated groups. *p*-values were calculated by two-sided Wilcoxon rank sum test. **i** UMAP plot shows NK cells subclusters identified. **j** Stacked bar graph shows the proportion of NK cells subtypes in siCont-LNA or siFXR1-LNA treated groups. *p*-values were calculated by two-sided Wilcoxon Rank Sum test. **k** Dot plots show the expression of signature genes selected for NK cell subtypes. Exact *p*-values for (**c**, **f**, **j**) are included in Source data file 1.

---

activation, and T cell mediated anti-tumor immune response in siFXR1-LNA group (Supplementary Fig. 9d).

Cell Chat analysis also identified other ligands and receptor interactions within M1/M2 macrophages and Cd8⁺ cytotoxic and Cd4⁺ cytotoxic T cell subtypes majorly through interaction with MHC-I/II signaling genes (such as *H2-d1, H2-aa, H2-k1, H2-dma, H2-m3* and *H2-t23*) in siFXR1-LNA treated group (Supplementary Fig. 9e and Supplementary Fig. 6c), as part of immune response mechanisms in siFXR1-LNA group. The cytokines and chemokines from cytotoxic Cd4⁺ and Cd8⁺ cytotoxic cells interact with their receptors (*Ccl12-Ccr2, Ccl2-Ccr2, Ccl7-Ccr2, Il18-Il18r1/Il18rap*) on M1-like macrophages were also identified in siFXR1-LNA group (Supplementary Fig. 9e), potentially for enhancing the immune response. Furthermore, interactions between *Cd80, Cd86* from macrophages to *Cd28* receptor present on cytotoxic T cells, again indicating a trigger for anti-tumor response by cytotoxic T cells in siFXR1-LNA treated group (Supplementary Fig. 9e and Supplementary Fig. 6c).

Natural Killer (NK) cells possess a unique capacity to directly engage and eliminate tumor cells, setting them apart from other immune cell types. Studies in both experimental mouse tumor models and in cancer patients have demonstrated crucial roles of NK cells in controlling tumor progression and inhibiting metastasis in patients[61–64]. Supporting this notion, treatment of siFXR1-LNA resulted in a notable increase in NK cells expressing *Ifng, Gzmb, Cd244a, Itga2, Il2rb,* and *Nkg7*, indicative of their cytotoxic and activated states against tumor cells (Fig. 6h, i). Our analysis of NK cell population further identified three distinct sub-clusters (Fig. 6j) defined by markers associated with cytotoxic and activated phenotypes. Among these, cluster 0 representing cytotoxic and activated form of NK cells was markedly enriched (~2-fold increase) in samples from siFXR1-LNA treated group (Fig. 6k). We also observed that cluster 1, representing a mixed-lineage population, and cluster 2, composed of partially cytotoxic and activated NK cells, were reduced in the siFXR1-LNA treated group (Fig. 6k). Altogether, our data highlights that siFXR1-LNA treatment enriched the cytotoxic and activated form of NK cells in vivo.

Our results demonstrate that the delivery of siFXR1-LNA encapsulated in jetPEI effectively evades RNase- and lysosome-mediated degradation, enabling efficient silencing of its target mRNA, FXR1, within tumor cells. Administration of the siFXR1-LNA complex also triggered multiple anti-tumor immune responses within the TME. Specifically, siFXR1-LNA treatment promoted the enrichment of Cd4⁺ and Cd8⁺ T cells, activated and cytotoxic NK cell subsets, and DCs with anti-tumor functions and suppressed the growth of tumor epithelial cells. Additionally, TAMs, which typically exhibit immunosuppressive phenotypes, were reprogrammed toward pro-inflammatory, anti-tumor phenotypes following siFXR1-LNA treatment (Fig.7).

## Discussion

Studies have demonstrated that FXR1 is an important gene for tumor formation, progression, and tumor relapse across various cancer types[9,65]. Our research has shown that FXR1 is an important oncogenic driver in ovarian cancer[9]. To be precise, FXR1 stabilizes *MYC* mRNA by binding to its AU-rich elements, then inducing mRNA circularization and recruiting translation initiation factors to the translation initiation site for facilitating MYC translation[9]. Previous research has demonstrated that down-regulation of another FXR1 family protein, FMRP, limits tumor growth in a mouse model of pancreatic ductal adenocarcinoma (PDAC) by influencing T cell inflammation and immunosuppression[66].

Therefore, our data suggest that FXR1 is a potential target for treatment for ovarian and other cancers that exhibit FXR1 amplification. However, its therapeutic potential has not been fully explored in translational research, primarily due to the lack of drugs that inhibit the actions of FXR1. Considering that there are no small molecule inhibitors or any other FDA approved agents readily available for FXR1 therapy, here we have developed LNA modified FXR1 siRNA as an RNAi therapy approaches for in vivo delivery to treat ovarian cancer as a model system.

Among the five different siRNA sequences targeting FXR1, the two most effective siRNAs with robust silencing of FXR1 expression, accompanied by notable suppression of tumor proliferation, colony formation, and cell cycle progression were selected for LNA structure modification. Importantly, our co-culture experiments revealed that siFXR1 selectively inhibited tumor cell growth without impacting normal epithelial cells. This selective action mitigates concerns about off-target effects and toxicity to normal cells, a common challenge in cancer therapeutics. The ability of siFXR1 to discriminate between tumor and non-tumor cells suggests a favorable therapeutic index, making it an attractive candidate for further development. Importantly, the incorporation of LNA modifications into the most effective siFXR1-3 significantly enhanced its efficacy, achieving over 90% inhibition of FXR1 expression at a concentration as low as 0.5 nM, compared to the 2 nM required by its native counterpart. This enhanced silencing translated into a more pronounced G1 phase arrest, superior inhibition of key cell cycle regulators, compared to the native siFXR1-3, along with similar level of induction in cellular apoptosis. These findings underscore the critical role of LNA modifications in enhancing siRNA potency and highlight siFXR1-3-LNA as a promising candidate for further therapeutic development. Mechanistic validation further revealed that tumors treated with LNA-modified siFXR1 showed a pronounced reduction in the proliferation marker Ki67 and an increase in apoptotic markers such as cleaved-caspase 3. These findings also highlight the potential of using LNA-modified siFXR1 as a targeted therapeutic approach for treating ovarian cancer and other cancers amplified FXR1.

In the past, we have used native forms of siRNAs and anti-miRs incorporated nanoparticles for treating the tumors[2,7–9]. However, we observed that the native form of small RNAs is unstable and exhibited low level presence in the recipient cells, which also warrants the use of LNA modifications in siRNA for in vivo delivery. As expected, siFXR1-

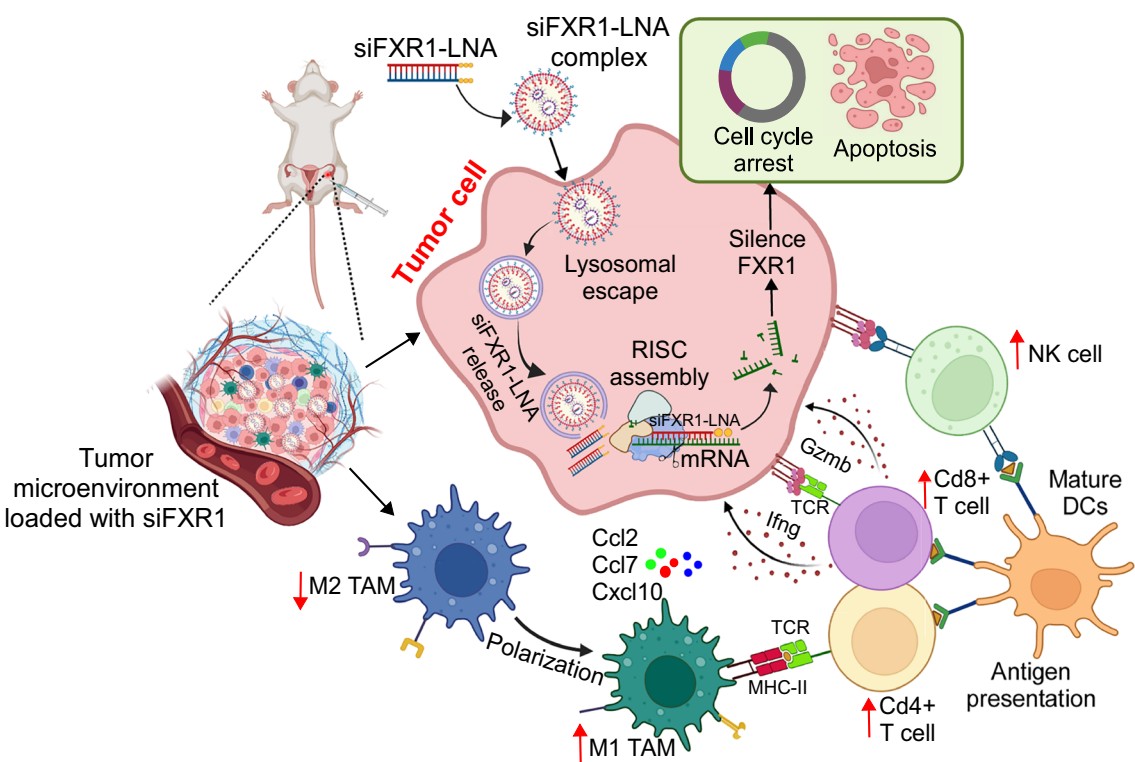

**Fig. 7 | Schematic model shows how the delivery of FXR1-siRNA-LNA in nanocomplex suppresses tumor growth, reprograms the immune microenvironment, and promotes immune cell activation.** The figure was partially created using BioRender image illustration software.

LNA exhibited improved uptake in ovarian cancer cells, with reduced localization in lysosomes that causes siRNA degradation, and improved stability from RNase and serum nucleases. Consequently, our siFXR1-LNA exhibited better tumor tissue uptake along with a correlative and improved tumor growth inhibition in both immuno-compromised and immunocompetent mice models in vivo. To therapeutically silence FXR1 in ovarian tumors in vivo, we used jetPEI nanoparticles incorporating the selected FXR1 siRNA. Pharmacodynamics of siRNA delivery systems are primarily influenced by the siRNA composition, where the choice of nanoparticles used for encapsulation can also affect these dynamics. Importantly, our choice of jetPEI nanoparticle used for the delivery of siFXR1-LNA did not exert any noticeable tissue damage due to organ toxicities, as evidenced by immunohistochemistry and biochemical assays of enzymes associated with liver and kidney functions.

Several RNAi-based approaches are currently undergoing assessment in phase I and II clinical trials[67]. However, the impact of siRNA sequences and the nanoparticles used for their delivery on immune cell modulation, immune cell suppression, or immune cell toxicity is not yet understood. The scRNA-seq technique has greatly advanced our understanding of tumor cells, tumor-associated stromal cells, and immune cells, particularly in terms of the genetic and molecular adaptations within tumor cells, thus providing deeper insights into the heterogeneity of tumor tissues. However, the impact of small molecule inhibitors, monoclonal antibodies, and RNAi agents on tumor progression using scRNA-seq technology was not well exploited in drug discovery research. Therefore, we employed scRNA-seq to characterize the impact of siFXR1-LNA in tumor cells, stromal cells, and immune cells in the TME with a goal of transitioning our preclinical testing to phase I and II clinical trials. Previous studies have utilized ascites samples as a clinically valuable source of transcriptional information, capturing tumor cells alongside tumor-associated immune and stromal populations. Ascites is particularly relevant in the context of advanced metastatic disease, where an immunosuppressive microenvironment predominates, enabling tumor cells to evade immune surveillance and contributing to therapeutic resistance.

Molecular phenotyping of epithelial cell clusters further revealed that siFXR1-LNA treatment markedly down-regulated the expression of cancer-associated epithelial cell markers and genes associated with cell cycle, proliferation, and translation. Cell Chat analysis[40] of cellular communication revealed that the control group expresses high levels of FXR1, displaying signaling tied to extracellular matrix components of collagens, laminins, and fibronectin that augment oncogenic signaling in tumor cells through autocrine mechanisms, as well as causing immunosuppressive pathways by abrogating immune cell activation. Actively growing tumors create an immunosuppressive TME, which is mainly manifested by the presence of TAMs, exhaustion of cytotoxic and helper T cells, lack of NK cell activity, presence of Tregs, and dysfunctional DCs responses. These factors are major obstacles to effective therapeutic interventions and contribute to the tumor's ability to evade the immune system. Macrophages in TME are classified into two polarized categories: M1-like and M2-like. M1-like macrophages, or classically activated macrophages, are known for their protective functions against cancer cell growth, whereas M2-like macrophages are considered key players in promoting cancer progression. Additionally, macrophages can impair T-cell functions and suppress their anti-tumor activity. Therefore, reducing the recruitment of TAMs, particularly M2-like macrophages, within the TME could help inhibit tumor growth. This mechanism was observed following our siFXR1-LNA treatment. Our flow cytometry analysis based on high levels of Pdl1 protein in M2-like macrophages in the control group demonstrates the mechanism of how ovarian cancer associated macrophages inhibit T cell function[68]. Conversely, M1-like macrophages characterized by upregulation of genes associated with anti-tumor responses and immune activation (e.g., *Ccl2, Ccl7, Cxcl10, Tgfb1*) were markedly enriched in the tumor site upon siFXR1-LNA treatment, suggesting a potential protective role of M1-like macrophages against

tumor growth. Moreover, such enrichment in macrophages in the FXR1 silenced group was associated with enhanced T cell activation within the TME.

The role of T cell populations within TME is a critical mechanism that directly influences tumor growth. Previous studies have shown that a low density of Cd8$^+$ cytotoxic T cells and Cd4$^+$ helper T cells, coupled with a higher prevalence of Tregs, is often associated with a worsened prognosis in ovarian cancer[69–71]. Using a combination of scRNA-seq, immunohistochemistry of tumors, and flow cytometry analysis, we observed an increase in both Cd4$^+$ and Cd8$^+$ T cells with cytotoxic characteristics as evidenced by high levels of *Nkg7, Prf1*, and *Gzmb* expression. In brief, the treatment with siFXR1-LNA led to a reduction in immunosuppressive T cell populations while enriching T cells characterized with anti-tumor properties, such as Cd4$^+$ and Cd8$^+$ T cells. This was further corroborated by an increase in NK cells expressing key activation and cytotoxicity markers, including *Ifng, Gzmb, Cd244a, Itga2, Il2rb*, and *Nkg7*.

It is expected that in a typical immune cell-activated setting, mature DCs initiate and sustain T cell-mediated anti-tumor immunity upon siFXR1-LNA treatment. The primary role of DCs is to prime naive T cells for proliferation (i.e., in antigen presentation), for which they are well equipped when mature[72]. In this study, we observed high expression of genes associated with DC maturation and migration, such as *Cd80* and *Cd86,* when treated with siFXR1-LNA. We observed that FXR1 knockdown modulates the DC landscape in the TME, shifting it from a pro-inflammatory to an anti-tumorigenic state, potentially enhancing anti-tumor immune responses. Specifically, the significant reduction in the ratio of MoDCs, which are associated with inflammatory responses and express markers such as *Cst3, Mrc1*, and *Arg1*, suggests a diminished recruitment of these pro-inflammatory cells to the TME. Conversely, we observed a notable increase in pDCs, characterized by the expression of *Bcl11a, Siglech, Bst2*, and *Ly6d*, which play a pivotal role in anti-tumor immunity. These cells not only promote the production of type I interferons (IFN-I) but also enhance the maturation of cDC1 expressing *Btla* and *Clec9a*, thereby boosting the cytotoxic functions of Cd8$^+$ T cells and NK cells. Previous studies by Färkkilä and colleagues have also demonstrated that chemotherapy remodels the immune TME of HGSOC by enhancing T-cell infiltration and promoting a polarization shift of macrophages from an M2 to an M1-like state, suggesting that therapeutic weakening of cancer cells can alleviate tumor-imposed immune suppression[73].

Overall, our single-cell profiling of ascites demonstrates that siFXR1-LNA therapy robustly suppresses tumor cell survival, proliferation, and immunosuppressive programs, while concomitantly enhancing anti-tumor immune responses. Although analyses of ascitic samples provide critical insights into the cellular and immune dynamics of ovarian cancer, they may not fully capture the spatial organization and contextual complexity of cell-cell interactions present within solid tumor tissues. To overcome this limitation, complementary approaches such as spatial transcriptomics applied to ovarian cancer solid tumors will be essential for delineating immune-tumor interactions within their native tissue architecture. Importantly, studies integrating spatial transcriptomic and spatial proteomic analyses with scRNA sequencing to precisely assess the cellular interactions are actively ongoing in our laboratory. Collectively, the data presented here establish a comprehensive mechanistic framework for our RNAi-based anti-tumor therapeutic strategy, achieved through in vivo delivery of siFXR1-LNA encapsulated in jetPEI nanoparticles. This highly efficient, tumor-targeted siRNA delivery system simultaneously inhibits tumor growth, reduces infiltration of tumor-promoting TAMs (M2-like macrophages), enriches tumor-inhibitory TAMs (M1-like macrophages), and enhances the presence of immunostimulatory T cells, NK cells, and dendritic cells. We envision that siFXR1-LNA represents a broadly applicable therapeutic modality for FXR1-overexpressing malignancies, and that nanoparticle-based delivery platforms offer powerful and versatile opportunities for advancing RNAi therapeutics in cancer treatment.

## Methods

### Ethics declarations
All animal studies performed in this study were approved by the Institutional Animal Care Committee (IACUC) at the Medical College of Wisconsin and conducted according to the committee's guidelines. Tumor growth/burden was monitored regularly, and mice were euthanized before tumors reached the maximum allowable size of 2 cm in maximal diameter recommended by our institutional ethics committee. Blood samples from humans were collected for PBMC isolation in de-identified manner with written informed consent in compliance with the institutional review board-approved protocol (PRO00041601) by the Ethics Committee of Froedtert Hospital and Medical College of Wisconsin. All samples were handled in compliance with the Declaration of Helsinki.

### Cell culture
HeyA8 cells were received from the Characterized Cell Line core at MD Anderson Cancer Center, Houston, TX, USA. Kuramochi cells were received from Taru Muranen at Beth Israel Deaconess Medical Center, Boston, MA, USA. OVCAR8 cells were purchased from the National Cancer Institute (NCI) cell line repository. NIH-OVCAR3 and MDA-MB-231 cells were purchased from the ATCC repository. Br-Luc and C-11 murine cell lines were received as a kind gift from Dr. Sandra Orsulic, University of California, Los Angeles, CA[37]. Ovarian surface epithelial (OSE) cells were isolated as described earlier[74]. FTE187 was received from Dr. Jinsong Liu at MD Anderson Cancer Center, Houston, Texas, USA. HeyA8, OVCAR8, Br-Luc, and MDA-MB-231 cell lines were cultured in DMEM medium (Sigma-Aldrich) supplemented with 10% fetal bovine serum (FBS, VWR), 1% Antibiotics (Anti-Anti 100X, Thermo Scientific). NIH-OVCAR-3 cells were maintained in the RPMI-1640 medium (ATCC, Manassas, VA, USA) supplemented with 10% fetal bovine serum (FBS, Omega Scientific),1% Antibiotics (Anti-Anti 100X, Thermo Scientific). OSE and FTE187 were maintained in cell culture medium consisting of 1:1 Medium 199 and MCDB-105 medium (Sigma-Aldrich) with 10% FBS, 1% antibiotics, 10 µg/mL Insulin Solution (Sigma-Aldrich), and 10 ng/mL hEGF (Peperotech). Cells were routinely tested and deemed free of PlasmotestTM Mycoplasma Detection Kit (InvivoGen, San Diego, CA). Authenticity of the cell lines used was confirmed by STR characterization at IDEXX Bioanalytics Services (Columbia, MO).

### Isolation of CD8$^+$ T cells from the PBMCs
T cells were isolated using the EasySep™ Human CD8$^+$ T Cell Isolation Kit (STEMCELL Technologies, Cambridge, MA, USA). Briefly, the frozen peripheral blood mononuclear cells (PBMCs) isolated from the whole blood were resuspended in the RPMI medium and centrifuged at 252 × g for 5 min. The resulting pellet was resuspended in RoboSep™ Buffer to which the Isolation Cocktail containing the antibody complexes was added at a concentration of 50 µL/mL of sample. This mixture was further subjected to negative selection by the addition of EasySep™ Dextran RapidSpheres. This resulted in the labelling of unwanted cells with antibodies and magnetic particles and the subsequent separation of the CD8$^+$ T cells, which were poured off into another tube and were used for further analysis.

### Western blotting
For preparing cell lysates, the cells were washed twice with 1x ice-cold PBS and lysed on ice in 1X RIPA lysis buffer containing freshly added 1x protease inhibitor cocktails (Sigma-Aldrich). For preparing the tissue lysates, the tumor tissues were homogenized in 1X RIPA lysis buffer over ice. After 30 min of incubation, the lysates were collected by centrifugation at 4 °C for 10 min at 11,180 × g. The amount of total

protein was determined using a BCA protein assay kit (Pierce, Rockford, IL, USA). An equal amount of total protein (30 μg) was resolved on precast 4–12% SDS-PAGE gels (Biorad, Hercules, CA, USA), transferred onto PVDF membranes, and incubated with desired primary antibodies, followed washing and incubation with HRP conjugated secondary antibodies (Cell Signaling Technology) and detecting of protein bands using chemiluminescence kit (Pierce, Rockford, IL, USA).

### siRNA transfection

All siRNA duplexes for human FXR1 and negative control were purchased from Integrated DNA Technologies, Inc. (IDT, Inc., Coralville, IA). siRNA sequences are listed in Table S1. Reverse transfections were performed using the Lipofectamine RNAiMAX transfection reagent (Thermo Fisher Scientific Inc., Waltham, MA). At 48 h post-transfection, cells were harvested for further analysis. Unlabeled and Texas red labeled LNA modified siRNAs were synthesized by Eurogentec (Liege, Belgium) and purified by high performance liquid chromatography (HPLC).

### Cell proliferation and colony formation assays

Cell viability was measured with the Cell Counting Kit- 8 (CCK-8) (Dojindo, Shanghai, China) according to the manufacturer's instructions. Cells were plated at a density of $1 \times 10^3$ cells per well in 96-well plates and incubated at 37 °C. Proliferation rates were determined at 24, 48, and 72 h post-transfection, and quantification was performed on a microtiter plate reader (Tecan, Mannedorf, Switzerland) at 450 nm.

For colony formation assay, transfected cells were plated in six-well plates at a density of 1000 cells per well. After 10 days, cells were rinsed with PBS, fixed in 5% glutaraldehyde for 10 min and then stained with 0.5% crystal violet (Sigma Aldrich, MO, USA) for 20 min. Plates were washed with water and dried before imaging.

### Cell cycle analysis

siFXR1s transfected ovarian cancer (Kuramochi and OVCAR8) cells were seeded at a density of $5 \times 10^5$ in 6-well plates. When cells reached 70–80% confluence, they were washed with PBS, trypsinized, collected, and fixed with 70% ethanol overnight. The next day, cells were treated with 1 mg/ml RNase A (Sigma-Aldrich, Saint Louis, MO) at 37 °C for 30 min and then resuspended in 0.5 ml of PBS and stained with 50 μg/ml propidium iodide (PI) (Sigma-Aldrich, Saint Louis, MO). The cells were analyzed using a FACScan flow cytometer (Becton–Dickinson, Mansfield, MA) and ModFit LT software (Verity Software, Topsham, ME).

### Annexin V/PI staining for apoptosis

Cellular apoptosis was measured with FITC Annexin V Apoptosis Detection Kit I (BD Pharmingen, San Diego, CA, USA) as per the manufacturer's protocol. In brief, Kuramochi and OVCAR8 cells were reverse-transfected with siRNAs (siCont and siFXR1s). After 48 h, the cells were trypsinized, washed with PBS, and resuspended in Annexin V binding buffer at a concentration of $10^6$ cells/ml. Annexin V– FITC (5 μl) was added, vortex-mixed gently, and incubated for 15 min at 4 °C in the dark. Cells were stained with 5 μl of PI for another 5 min at 4 °C in the dark. Stained cells were acquired on a FACS Calibur flow cytometer (Becton–Dickinson, Mansfield, MA) and data were analyzed with Flowjo software version 10.6.1 (TreeStar, Ashland, OR, USA).

### Caspase3/7 activity assay

Following treatments with siFXR1s for 48 h, cells were subjected to Caspase 3/7 activity measurement with Caspase-Glo assay kit (Promega, Madison, USA). Briefly, the plates containing cells were removed from the incubator and allowed to equilibrate to room temperature for 30 min. 100 μl of Caspase-Glo reagent was added to each well, and the contents of well was gently mixed with a plate shaker at $28 \times g$ for 30 s. The plate was then incubated at room temperature for 2 h. The luminescence of each sample was measured in SpectraMax i3x Multi-Mode Microplate Reader (Molecular Devices, Japan Co., Ltd., Tokyo, Japan).

### Cellular uptake and lysosomal escape of siRNA

To examine the uptake and escape of the siRNAs from lysosomes, native siFXR1 and siFXR1-LNA labelled with Texas-red, kuramochi and OVCAR8 cells were seeded in the 35 mm glass-bottom tissue culture plates (Cellvis LLC) and allowed to attach overnight. On the second day, cells were transfected with respective siRNAs using Lipofectamine RNAiMAX transfection reagent (Thermo Fisher Scientific Inc). Cells were washed twice with 1x PBS and stained with Hoechst 33342 and LYSO-ID® Green detection kit (Enzo Life Sciences, New York, USA) before imaging. Images of the live cells were captured at 1, 3, 6, and 24 h post siRNA treatment with a 40X objective using a confocal laser scanning microscope (LSM 510; Zeiss, Oberkochen, Germany).

At 48 h of siRNAs treatment, a TUNEL assay was conducted using an In Situ Cell Death Detection Kit (Roche Applied Science, IN) according to the instructions to detect apoptotic cells. Images from each treatment group were evaluated using a confocal laser scanning microscope (LSM 510; Zeiss, Oberkochen, Germany), and TUNEL-positive cells were quantified using Aim 4.2 software, LSM 510.

### Serum and enzymatic stability assays

To study the resistance to RNase I, native siFXR1 and LNA modified siFXR1 complexed with JetPEI containing a total amount of 1 μM siRNA were incubated with 1 unit of RNase I at 37 °C for 30 min, followed by incubation at 90 °C for 30 min to inactivate the enzyme. Heparin (200 μg mL$^{-1}$) was added to the samples, and the mixture was incubated for an additional 30 min at room temperature to release the siRNA from the lipid complex. 1% TAE-Agarose gel electrophoresis was used to determine siRNA integrity, and siRNA band intensity was quantified using ImageJ software.

Serum stability study was performed with human and mouse sera as previously reported[9]. The native siFXR1 and LNA modified siFXR1 complexed with JetPEI incubated with 10% human or mouse serum for different time intervals at 37 °C. siRNAs were then released from the complex by incubation with heparin (200 μg mL$^{-1}$) for 30 min and then analyzed by electrophoresis in a 20% TBE-polyacrylamide gel electrophoresis.

### Live cell incucyte assay

OSE and FTE cells were nuclear-labeled red with the Incucyte® Nuclight Rapid Red Dye for Live-Cell Nuclear Labeling (Sartorius, USA) according to the manufacturer's protocol. GFP labelled OVCAR8 were individually co-cultured with OSE and FTE NucLight Red cells in 1:1 ratio and allowed to attach overnight. The media were removed, and cells were transfected with siCont-LNA and siFXR1-3-LNA using Lipofectamine RNAiMAX transfection reagent. After 24 h, cocultured cells were trypsinized, and $5 \times 10^3$ cells were reseeded in quadruplicates on a 96-well plate containing complete medium. Live images were taken every 6 h for a period of 72 h using the IncuCyte® Live Cell imaging system (Sartorius, USA). Green cell counts based on number of live cells were performed using IncuCyte® S3 Software (Sartorius, USA).

### Co-culture assay

Twenty-four hours prior to transfection with siCont-LNA or siFXR1-3-LNA, OSE and FTE cells and GFP expressing OVCAR8 cells were seeded together into a 6-well, glass bottom slides (Cellvis LLC) plate at 1:1 ratio, respectively. Prior to imaging, cells were stained with Hoechst (Life Technologies; 5 ng/mL). Plates were then imaged at day 0 (day of transfection), 2, 4, and 6 days after transfection using a 40X objective using a confocal laser scanning microscope (LSM 510; Zeiss, Oberkochen, Germany). Counting on the GFP channel was done using the Aim

4.2 software, LSM 510, and used to quantify the number of OVCAR8 cells in the culture. Total cell populations were quantified by counting Hoechst+ nuclei using the Aim 4.2 software LSM 510. The proportion of OVCAR8 cells to total cells in each well was quantified by dividing the number of GFP+ cells by total Hoechst+ cells.

## Comet assay

To evaluate DNA damage in a cell, we performed an alkaline comet assay using Comet SCGE assay kit according to the manufacturer's instructions (Enzo Life Sciences, New York, USA). Briefly, Kuramochi and OVCAR8 cells were transfected with siCont, siFXR1-3, and siFXR1-3-LNA using Lipofectamine RNAiMAX transfection reagent. After 48 h, cells were gently scraped and suspended at $1 \times 10^5$ cells/mL in ice cold PBS. 50 μL of cells combined with molten LM Agarose at a ratio of 1:10 (v/v) were dropped and spread onto Comet Slide. After gelling LM Agarose, slides were treated with lysis solution for 1 h at 4 °C and alkaline unwinding solution for 20 min at room temperature. The slides were placed in electrophoresis tray containing TBE electrophoresis solution and subjected to electrophoresis at 21 volts for 15 min. After staining cells with CYGREEN green Nucleic acid dye, cell images were obtained at 20X magnification using EVOS M5000 imaging system (Thermo Fisher Scientific). Percent DNA in the tail and tail moment or comet-like structures from at least 50 randomly selected cells per sample were analyzed using OpenComet v1.3.1 software[75].

## In vivo study

We performed sequence alignment analysis for both siFXR1 sequences (seq2 and seq3) using CLUSTALW before using them for the in vivo study. This analysis revealed that seq3 has 96% sequence homology between human (NCBI Reference Sequence: NM_001013439.3) and mouse (NCBI Reference Sequence: NM_008053.4) FXR1, supporting its use in the murine model.

Athymic nude mice (J:NU, Strain #007850) and FVB/NJ mice (Strain #001800), female and approximately 4–6 weeks old, were purchased from Jackson Laboratories. Ovarian cancer affects only females; therefore, only female mice were used in this study. The animals were maintained under a 12 h light/dark cycle with unrestricted access to food and water at the temperature around 20–24 °C with 40–60% humidity. For the tumorigenicity study, OVCAR8 ovarian cancer cells expressing luciferase ($0.5 \times 10^6$ cells/mouse) were injected IP into mice ($n = 7$ per group) to establish a human ovarian cancer xenograft model. After 7 days, tumor-bearing mice were randomized into three groups and treated IP with 10 μg/mouse of siCont-LNA, siFXR1-LNA, or siFXR1, each complexed with 16 μL of in vivo-jetPEI transfection reagent (PolyPlus Transfection, Illkirch, France) in a final volume of 200 μL of 5% glucose. Treatments were administered twice weekly for 28 days[76]. Mice were regularly monitored for tumor growth by bioluminescence imaging. For bioluminescence imaging, mice were IP injected with 150 mg/kg of D-luciferin in 200 μL PBS, and bioluminescence signal was captured using an IVIS Lumina II in vivo imaging System (Caliper Life Sciences).

To evaluate siRNA toxicity, athymic nude mice were randomly assigned to three groups ($n = 5$ per group) and administered IP injections twice weekly for 4 weeks with 10 μg/mouse of siCont-LNA, native siFXR1, or siFXR1-LNA, each formulated with jetPEI nanoparticles. After the final injection, major organs (kidney, lungs, liver, and brain) were collected, fixed, sectioned, and stained with H&E for histopathological assessment. Blood serum was also collected and submitted to IDEXX BioAnalytics (Columbia, MO) for hematological and biochemical analyses, including measurements of ALT, AST, bilirubin, albumin, and creatinine.

For the breast tumorigenicity study, female athymic nude mice were injected subcutaneously with MDA-MB-231 cells ($5 \times 10^6$ cells/mouse) mixed 1:1 with Matrigel (Corning). Once tumors reached

70–100 mm³, mice were randomized into three groups ($n = 5$ per group) and treated via retro-orbital injection with 10 μg/mouse of siCont-LNA, siFXR1, or siFXR1-LNA, each complexed with 16 μL of in vivo-jetPEI transfection reagent (PolyPlus Transfection, Illkirch, France) in 100 μL of 5% glucose. Treatments were given twice weekly for 7 weeks. Tumor size was measured weekly using a vernier caliper, and tumor volume was calculated as: volume (mm³) = length × (width²)/2. Mice were euthanized at week 7.

For the scRNA-seq study, FVB/NJ mice were IP injected with $0.5 \times 10^6$ Br-Luc cells. Seven days after tumor inoculation, mice received IP injections of siCont-LNA ($n = 10$) or siFXR1-LNA ($n = 10$), the latter designed to target the mouse FXR1 sequence (NCBI Reference Sequence: NM_008053.4), each formulated with jetPEI as described above. For the survival study, mice exhibiting moribund conditions with ascites formation were considered to have reached the experimental endpoint and were euthanized. Ascites and tumor tissues were collected from each group for downstream analyses.

## Bioluminescence imaging

In vivo bioluminescent imaging was performed using the IVIS Lumina II Bioluminescence and Fluorescence Imaging System (Caliper Life Sciences). Mice were injected D-luciferin (150 mg/kg; Gold Biotechnology, St. Louis, MO) intraperitoneally (IP), and imaging was conducted 10 min later, corresponding to the peak signal. Tumor images were captured under the following settings: exposure time, 0.5 s; f/stop, 16; medium binning; and a 12.5 × 12.5 cm² field of view. Bioluminescent signals were quantified using Living Image software and reported as tissue radiance (photons/s/cm²/sr).

For the in vivo uptake and biodistribution study, tumor-bearing mice were imaged using the IVIS Spectrum System (PerkinElmer, Shelton) at 24 h after a single IP injection of Texas Red–labeled siFXR1-LNA. Following imaging, mice were euthanized, and tumors and major organs were collected and imaged to assess siRNA distribution.

## Immunohistochemistry (IHC) and TUNEL assay

For this purpose, the slides were dewaxed in xylene and rehydrated through graded ethanol to distilled water. Antigen retrieval for the slide specimens was performed using IHC-Tek epitope retrieval solution and steamer set (IHC World, LLC.). The slides were then immersed in 3% $H_2O_2$ for 10 min to quench endogenous peroxidase, followed by blocking with 10% goat serum for 1 h. Tissue sections were then stained overnight at 4 °C with the indicated primary antibodies. The slides were counterstained with Harris modified hematoxylin (Thermo Fisher Scientific Inc., Rockford, IL), dehydrated with graded ethanol and xylene, and finally mounted with Paramount. IHC slides were then digitally scanned using Pannoramic 250 FLASH III scanner (3D HISTECH ltd. Version 2.0), using the Case Viewer software (3D HISTECH ltd. Version 2.0) was used to view the images.

A TUNEL assay kit (Roche, Basel, Switzerland) was used to determine apoptotic cells in tumor specimen according to manufacturer's instructions. Images from each treatment group were evaluated using a confocal laser scanning microscope (LSM 510; Zeiss, Oberkochen, Germany), and TUNEL-positive cells were quantified using Aim 4.2 software LSM 510.

## 3D tumor spheroid invasion and penetration study

3D tumor spheroids were prepared with growth factor reduced matrigel (GFR Matrigel, Corning Life Sciences, NY, USA) as previously reported. Thousand OVACR8 cells were suspended in 100 μL of spheroid formation ECM and added into a Corning 96-well ultralow attachment microplate, followed by centrifugation at $200 \times g$ for 3 min at 4 °C. The plate was incubated at 37 °C for 24 h to induce the formation of spheroids. The tumor spheroids were incubated with siFXR1-LNA labelled with Texas-red to evaluate the tumor penetration capability of siRNA.

For spheroid invasion assay, the spheroids were embedded in matrix from the Cultrex Spheroid Cell Invasion Assay kit (Trivigen, Gaithersburg, MD) at 37 °C for 60 min and transfected with the siCont-LNA and siFXR1-LNA using Lipofectamine RNAiMAX transfection reagent for 24 h. The medium was then replaced by DMEM containing 10% FBS with 5 ng/mL hEGF, and the spheroids were incubated at 37 °C for up to 4 days. Invasion of the spheroids in the matrix was photographed every 24 h using an inverted microscope (Nikon Ti2E).

## scRNA-seq sample preparation, data acquisition, and pre-processing

Ascites collected from three biological replicates were pooled together as one sample, and a set of two samples per treatment group (siCont-LNA or siFXR1-LNA) was used for scRNA sequencing. Samples were then processed for single cell using tissue dissociation enzymes. scRNA-seq was performed using the Chromium Next GEM Single Cell 3′ Reagent Kits v3.1 (Dual Index). Briefly, cells were loaded into the 10× Chromium Controller (10× Genomics) for barcoding. scRNA-seq libraries were then generated according to the manufacturer's protocol. After the scRNA-seq library construction, the Agilent 4200 Tape Station system and High Sensitivity D5000 and D1000 Screen Tapes were used to assess the size profiles of the amplified cDNA. A NextSeq 500/550 High Output Kit v2.5 (150 cycles; 20,024,907; Illumina) with 28 cycles for read 1, 10 cycles for i7 index, 10 cycles for i5 index, and 90 cycles for read 2, and was used to sequence the samples. Raw sequencing data were downloaded from Illumina BaseSpace, then demultiplexed and converted to gene-barcode matrices using the "mkfastq" and "count" functions in Cell Ranger v8.0 (10× Genomics). To reduce potential batch effects among samples, all cDNA libraries were constructed using the same reagent kit and protocol.

## scRNA-seq data analysis for cell type clustering and annotation

After aligning raw sequencing reads to mouse mm10 reference genome, cell barcode and unique molecular index (UMI) count matrices were generated. Pre-processing of the data and downstream analyses were performed in R (v.4.3.3) using Seurat package (v.5.1.0). Cells expressing less than 100 or higher than 4800 genes or expressing higher than 15% of UMI counts from mitochondrial genes were removed due to poor quality or potential doubles. After quality control, a total of 16,851 cells remained for downstream analysis. We used log2 transformation with 10,000 as scaling factor for normalization, and then selected the top 2000 most variable genes, followed by performing principal components analysis (PCA) for denoising the data into 50 PCs.

During our analysis, we found that one of the replicates among the siFXR1-LNA group exhibited fewer sequencing reads, which resulted into identifying less cell numbers, presumably due to the therapeutic effects of siFXR1 LNA and associated decrease in overall tumor burden and ascites (Supplementary Fig. 5e, f). This sample was excluded for further analysis. In brief, we identified 16851 high-quality cells (~11621 cells from siCont-LNA and ~5230 cells from siFXR1-LNA group) for further subtyping (Supplementary Fig. 5h).

Shared nearest neighbor (SNN) of each cell was constructed for each cell and then clustered using the Louvain-Jaccard graph-based algorithm, implemented by functions FindNeighbours and FindClusters with the resolution parameter set to 0.1. We used Uniform Manifold Approximation and Projection (UMAP) as dimensional reduction algorithm to visualize the data in two-dimension along with cluster results from the clustering algorithm. Cluster marker genes were identified by using Wilcoxon rank sum tests with adjusted $p$-value less than 0.05 (with Bonferroni correction for multiple testing). The main cell types were defined using cluster marker genes along with canonical markers.

Dimension reduction plots, heatmaps, violin plots, and dot plots were generated using functions DimPlot, DoHeatmap, VlnPlot, and DotPlot provided in Seurat, respectively. Bar plots were made by using the R package ggplot2 (version 3.5.1).

The scoring of cell function based on a specific gene set was performed through the built-in "AddModuleScore" function of the Seurat package.

## Subset analysis for epithelial, macrophage, T, NK, and Dendritic cells

Specific subgroup of cells was extracted from original dataset and re-ran the whole analysis pipeline as aforementioned, including normalization, select highly variable genes, denoising procedure by PCA, clustering analysis, and dimension reduction by UMAP. We performed pathway enrichment analysis with the differentially expressed gene list (Supplementary Data 1) from each epithelial cell cluster using the IPA software.

## CNV estimation and cell cycle analysis

To identify malignant cells from epithelial cells, we used the R package CopyKAT (version 1.1.0)[42] to estimate the CNVs for each cell and predict if it is aneuploid or diploid. We assessed the cell cycle phase of the cells using the CellCycleScoring function[77] provided by the Seurat package.

## Cell–cell communication analysis

The R package CellChat (v2.1.2)[40] was used to analyze cell-to-cell communication between tumor cells and other cell types. First, a CellChat object was created by grouping defined clusters. The ligand–receptor interaction database we used for analysis was "CellChatDB.mouse", without additional supplementation. Preprocessing steps were all conducted with default parameters. The functions computeCommunProb and computeCommunProbPathway were applied to infer the network of each ligand–receptor pair and each signaling pathway separately. A circle plot was also generated as part of this analysis. The number of interactions and strength plots were made by using netVisual_diffInteraction function.

## Flow cytometry of ascites, TAM, and T cells

After blocking with Fc Receptor Blocking Solution (Biolegend), cell surface staining was performed in FACS buffer containing antibody cocktails (Epcam, Cd45, F4/80, Cd11b, Pdl1, Mrc1 (Cd206), Cd163, Cd4, Cd8) on ice for 1 h. After washing twice with FACS buffer, the cells were fixed using BD Cytofix/cytoperm solution (BD Bioscience) 20 min on ice. Cells were washed with BD perm/wash buffer (BD Bioscience), intracellular blocking with mouse IgG and Miltenyi FcR blocking, then intracellular staining was performed using PE-anti-LAMP3 (BD Biosciences), or PE-isotype control (BD Biosciences) for 1 h on ice. Cells were washed twice with perm/wash buffer and then analyzed on the BD Fortessa.

## Tissue IF staining

FVB mouse tumor tissue slices from siCont and siFXR1-LNA groups were dewaxed in xylene and rehydrated through graded ethanol and distilled water. Antigen retrieval for the slide specimens was performed using IHC-Tek epitope retrieval solution and steamer set (IHC World, LLC.). The slides were then immersed in 3% $H_2O_2$ for 10 min to quench endogenous peroxidase, followed by blocking with 10% BSA for 1 h. Tissue sections were then stained overnight at 4 °C with the indicated primary antibodies. After washing with PBS, tissue sections were incubated with secondary antibodies, Alexa Fluor goat anti-mouse 488 (Cat#38731, Life Technologies, Carlsbad, CA) and Alexa Fluor 568 goat anti-rabbit (Cat#35646, Life Technologies, Carlsbad, CA) for 1 h at room temperature. Glass slides were mounted using ProLong Gold Antifade Reagent (Life Technologies, Carlsbad, CA) containing DAPI. Images were acquired with a 40X objective using a confocal laser scanning microscope (LSM

510; Zeiss, Oberkochen, Germany) and analyzed using the Aim 4.2 software LSM510.

## Statistical analysis

In most cases, data obtained from three or four biological replicates were analyzed, unless indicated otherwise in the figure legends. The statistical tests used for each experiment are included in the corresponding figure legends. Graphpad Prism 7 (GraphPad, San Diego, CA) was used to perform statistical analysis and *p*-value determinations.

## Reporting summary

Further information on research design is available in the Nature Portfolio Reporting Summary linked to this article.

## Data availability

The raw scRNA-seq data generated in this study are available through the Gene Expression Omnibus (GEO) under accession number GSE292799.

The remaining data are available either within the article, Supplementary Information or Source Data file. All source data files are provided with this paper. Source data are provided with this paper.

## Code availability

The analysis in this study was performed using standard and publicly available R packages as detailed in the Methods section. No custom codes were developed for any analyses in this study. Therefore, no additional code is available for public sharing.

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

## Acknowledgements

This work was supported in part by grants to P.C.-R. from DoD W81XWH-21-1-0365, HT9425-23-1-0311, 1R01CA291708-01A1, and Linda G. and Herbert J. Buchsbaum Endowment, Women's Health Research Program (WHRP) funds, and the Sharon L. La Macchia Innovation Fund at MCW. S.P. was supported by the Department of Defense (DoD W81XWH-21-1-0361 and W81XWH-21-1-0138) and NCI R01CA258433. J.G was partially supported by MCW Cancer Center postdoctoral award. P.C.-R. was also supported by seed funds from MCW Cancer Center and Advancing Healthier Wisconsin Endowment funds. We also thank the animal core facility, and biomedical imaging shared resource core of Medical

College of Wisconsin, and the Children's Research Institute's histology core. All contributing authors reviewed and provided consent for publication.

## Author contributions
P.C.R. conceived the study, generated hypotheses, designed experiments, and analyzed the results. J.G. conceived the study, designed and performed most of the experiments, including cell cultures, animal experiments, microscopy, immunoblots, statistical analyses, prepared figures, and the draft of the manuscript. X.M., A.N., S-W.T., and C-W. L. performed all the bioinformatics and computational analysis for this study. I.P.K., S. M., S.K., M.S., M.P., A.G., and Anupama. N. assisted with animal experiments, animal imaging, in vitro experiments, or biochemical assays. J.M.J. assisted in pathological analysis. S.P. provided feedback on animal experiments and assisted with manuscript preparation. A.N., S. M., F.D., M.P., and C-W. L edited the manuscript and provided comments. E.H. and S.M. assisted with scRNA-seq experiments. P.C.R. provided scientific direction, established collaborations, prepared the manuscript with J.G., and allocated funding for the work.

## Competing interests
P.C.-R, S.P., and J.G. are inventors of a US provisional patent application 63/683,329 entitled compositions targeting FXR1 and methods of using the same for the treatment of diseases and disorders associated with FXR1 expression. The remaining authors declare no competing interests.
