## [Peer Review File · Nature Communications]

Single-Cell Transcriptomics Reveals FXR1 as an Actionable Target for siRNA Therapy in Ovarian Cancer

Corresponding Author: Professor Pradeep Chaluvaly-Raghavan

Version 1:

Reviewer comments:

Reviewer #1

(Remarks to the Author)

The authors demonstrated that FXR1 is a promising therapeutic target for the elimination of ovarian cancer cells. They showed that siRNA-mediated silencing of FXR1 selectively induces apoptosis and G1 phase cell cycle arrest in ovarian cancer cells, but not in normal ovarian surface epithelial (OSE) and fallopian tube epithelial (FTE) cells—likely due to the relatively low expression of FXR1 in these normal cells. To enhance therapeutic efficacy, the authors incorporated locked nucleic acid (LNA) modifications and polyethylenimine (PEI) encapsulation of the siRNAs, which improved their stability, cellular and tissue uptake, and overall in vivo therapeutic performance. Finally, using single-cell RNA sequencing (scRNA-seq), they demonstrated that delivery of LNA-modified FXR1 siRNAs in mice triggered an anti-tumor immune response, characterized by tumor-associated macrophage (TAM) polarization and increased infiltration of dendritic cells (DCs), natural killer (NK) cells, and T cells.

Although the authors have performed extensive analyses of the effects of siFXR1 both in vitro and in vivo, the storyline is difficult to follow as it drowns to the vast amount of oftentimes unjustified or redundant analyses shown in 9 main and 10 supplementary figures. In addition, the choice of methods and models are not consistently justified, statistics are lacking, claims are not always supported by the data shown, and the novelty of results shown in first figures is limited. Hence, we do not recommend the manuscript for publication in Nature Communications. We delineate our criticism below and hope the authors can use these to improve their manuscript for following submission(s) to another journal.

1. Introduction does not sufficiently refer to previous work on using siRNA modifications in cancer research, including in vivo models and even clinical trials (see e.g., reviews doi.org/10.1038/s41392-020-0207-x, doi.org/10.1016/j.jconrel.2023.07.054). This leads to overstatement of the novelty of the conceptual idea behind this article.

2. The part of FXR1 siRNA selectively inducing apoptosis and G1 phase cell cycle arrest in ovarian cancer cells is clear, but the novelty is compromised since the authors have reported similar phenotypes previously (ref 9).

3. As stated above, the current form of nine figures is laborious for the reader and way too lengthy for the relatively simple message of the manuscript. The core data of Figures 1–3 on the optimisation of the siRNA could be combined to one figure, and the scRNA-seq analysis in Figures 5–9 to one to two figures.

4. The advantages of LNA modification and PEI encapsulation of siRNA appear promising, particularly in enhancing stability and delivery. However, the in vivo results were not fully convincing, as the native FXR1 siRNA alone already led to a significant reduction in tumor burden, making it difficult to clearly attribute additional therapeutic benefit to the modifications.

5. There are several results that lack sufficient statistical analyses, and hence conclusions presented based on them are not justified by the data. For example, Fig. 1c, and Fig. 3g and Fig. S4c lack quantification and statistical analyses; hence, the claims made on these data are not justified. Also, e.g., Figs. 5d, 8c, and 9f lack error bars and statistics (the data should contain two repeats according to Fig. 5a).

6. The mouse models containing KRAS^{G12D} are not optimal, as KRAS mutations are extremely rare in HGSC. Also, the authors assessed two mouse cell lines, one with and one without MYC amplification, but chose the cell line without MYC amplification for in vivo experiments without any justification. Importantly, the authors use FXR1's role in MYC translation as a justification for its mechanistic importance in tumorigenesis but do not address it at all, even though they have a syngeneic pair of cell lines with and without MYC to directly do this.

7. Throughout the manuscript, the authors make unjustified claims based on CellChat analysis. CellChat results do not “determine the ligand-receptor interactions”; they present potential for interactions. CellChat results can neither reveal mechanisms but rather potential mechanisms.

8. The use of CIBERSORT, meant for bulk RNA-seq signal deconvolution, to what authors state as “a more detailed characterization” when analysing differences in cell-type signature scores between specimens is not well justified. There are many widely used scoring methods—such as AUCell, ssGSEA, AddModuleScore—for this exact purpose.

9. The scRNA-seq analyses of Figure 6 are mostly obscure. For example, why do the authors calculate “overall tumor scores” to differentiate between tumor versus normal cells, when they have CopyKAT data (or can perform inferCNV) to robustly separate tumor cells from normal cells?

10. From the scRNA-seq data, the authors mentioned that there were about 3500 epithelial cells (Fig. S5h). Among them, about 3000 cells from siCont-LNA and 500 from siFXR1-LNA (Fig. 5d). But from lines 415–419, “highest numbers of aneuploid epithelial cells (5400) over normal epithelial cells (120 diploid cells)”, the cell number of epithelial cells is very unclear and does not match to Fig. 6a. And there were almost no tumor epithelial cells at all from siFXR1-LNA group, making us to wonder the sense of CellChat analysis between normal rather than tumor epithelial cells and TME.

11. The author designed 5 siRNAs to target human FXR1, of which seq2 and seq3 were demonstrated with high efficiency. Later, the author used them (seq2 and seq3) also for targeting mouse FXR1, achieving good knockdown efficiency (Fig. S5a). However, the BLAST results seem they (seq2 and seq3) are not 100% matching to mouse FXR1. Have the authors checked the potential off-targets when using human FXR1 siRNAs for mouse FXR1?

12. For the in vivo experiments, it would be helpful to test a less frequent dosing schedule (e.g., extended intervals between siRNA administrations) to more clearly demonstrate the advantage of using LNA-modified siRNAs over the native form. This would better reflect the improved stability and durability that LNA modifications are intended to provide.

13. For scRNA-seq experiments, why the author chose ascites and not the tumor mass? This should be clarified.

Minor Comments

15. The manuscript requires a proper language check.

16. Abstract: “In tumor microenvironment (TME), siFXR1-LNA diminished tumor cell proliferation” should read “In vivo, siFXR1-LNA diminished tumor cell proliferation ...”

17. Abstract: “such as copy number gain, amplification, or deletion of genome.” should be corrected to “...deletion of genomic elements.”

18. The ‘native’ siRNAs (siFXR1-2, siFXR1-3) have significant effects on cell cycle in Fig. 1b but not in Fig. 2d; please explain this inconsistency.

19. It is not clear what is the difference between Fig. 3g and Fig. S4c. Also, as referred to above, the result text states in both that tissues with tumor have more Texas Red signal from the siRNA, but the amount of tumor is not shown or quantified.

20. Fig. 4g: how can the proportion of TUNEL-positive cells be over 100%? Doesn't this suggest that some of the TUNEL signal is artifactual?

21. Line 358–361: the authors said “the interactions were notably higher in the control group”, is that true?

22. Line 412: there is no “Fig. S5k”.

23. Fig. 5d y-axis label is marked “%”, which does not match the actual data. A similar problem appears in Fig. 8f.

24. Fig. S3b flow cytometry figure is not properly presented for visualization.

25. In Figure S7a, the x-axis labels “0, 1, 2” are unclear — could the authors clarify what these values represent?

26. “Therefore, FXR1 is considered as an excellent molecular and actionable target for ovarian and other cancer therapy.” Isn’t this a bit overstatement? One would assume that excellent targets are targetable by small molecules.

Reviewer #2

(Remarks to the Author)

This is a nicely executed study demonstrating the therapeutic potential of a locked nucleic acid (LNA)-based siRNA targeting FXR1 for ovarian cancer treatment. The engineered siFXR1-LNA showed enhanced stability against RNase degradation, improved tumor uptake, and potent suppression of FXR1 expression compared to native siRNA. Its delivery via polyethylenimine significantly reduced tumor growth and peritoneal metastasis in both nude and immunocompetent mouse models, without observable toxicity. Single-cell RNA sequencing revealed not only direct inhibition of FXR1 in cancer cells but also disruption of oncogenic translation pathways and normalizing the tumor microenvironment’s immune function. A range of complementary methods supports the authors’ conclusions, which are well-reasoned and convincing.

Critique:

Please state whether sc RNA-seq dataset is available and the data bank where it is deposited.

Please comment on why the single-cell analysis was done with ascites and not tumors. The benefit of using ascites is less variability in cell types, but the benefit of tumors is the spatial context, which is crucial for cell signaling in immune infiltrates. A good part of the work was the description of cell-cell signaling inference by CellChat. Is it assumed that even after these immune cells shed into the ascites, they retain the transcripts of immune activation signaling?

Lane 413 and Fig. 6 a, c, and h: I am confused by the definition of tumor cells and nonmalignant epithelial cells. I understand that this distinction was made using CopyKAT CMV inference. Based on the CNV inference, most of the cells in the ascites are nonmalignant cells. What are these epithelial cells if they are not tumor cells? Mesothelial cells?

Fig. 4h: It appears that there is more blood in FXR1-treated lungs compared to the control. Is this by chance in the selected photographs or there is actually more blood when whole lungs are examined?

The three figures below could benefit from a panel showing H&E staining, which would clarify some of the questions below:

Fig. 4f: Two of the panels stained with Ki67 seem to have high background – all of the cells are positive, which is unlikely even in very aggressive tumors.

Fig. 7g: There are two sets of staining shown for both control and FXR1-treated mice. It is unclear whether these represent two distinct tumors per treatment group. The nuclear staining does not clearly identify the tissue context—are these epithelial tumor cells or immune infiltrates? In the FXR1-treated group, the left panels appear to show necrotic debris rather than intact nuclei, while the right panels suggest an area of immune cell infiltration. In contrast, the control panels appear to represent tumor tissue. Additionally, the F4/80 staining appears nonspecific.

Fig. 8e: The panels showing tumors from control mice have different hematoxylin staining from the siFXR1-LNA-treated mice. Do these sections represent tumors or necrotic cells?

Reviewer #3

(Remarks to the Author)

Reviewer #4

(Remarks to the Author)

This is an outstanding and very innovative study,

Version 2:

Reviewer comments:

Reviewer #1

(Remarks to the Author)

The authors have improved the paper but still the issues are such that I cannot recommend the publication of the manuscript in Nature Communications.

A major weakness of the paper is the conclusions drawn from the scRNA-seq analysis. The analysis is displayed still very extensively in Figures 4 to 6, with conclusions used in Fig. 7. Yet, it is mostly irrelevant since it is comparing tumor microenvironment to a non-tumor microenvironment, as the authors state that the treated group has only 2 aneuploid (=cancer) cells. That means that based on the data shown, one cannot specify which of the observed changes in for example immune interactions are due to the FXR knockdown, and which are simply due to the loss of cancer. Here, the results are presented and concluded upon as if they would inform us of the effects of loss of FXR1 expression, which is misleading.

Overall, the effect of the modified siRNA on tumor growth is clearly shown in the mouse model used, with also convincing evidence of how the stabilising modification improves the response when dosing is more sparse. Also the effect on Myc expression is now clear. However, these more solid aspects form just a half of the manuscript.

Response to comment 11:

The match of seq3 is not 100% as the authors state in their response. The authors fail to highlight the "AGA" at the end of the sequence in their response, although it is present in Table S1: "Seq3: GCUUACUUGAUAAUACAGAAUCAGA". Hence, their response is incorrect.

Minor:

Response to comment 6.

The authors now state that the model used, Br-Luc does not have KRAS mutation, even though their previous manuscript version that was reviewed stated on rows 311-312: Br-Luc (genotype: p53^{-/-}; BRCA1^{-/-}; myc; KrasG12D; Akt-myr) and C11 (genotype: p53^{-/-}; BRCA1^{-/-}; myc; KrasG12D). If there was an error in this version the authors could have simply acknowledged that there was an error.

Response to comment 11:

In addition to the misleading highlighting of the target sequence, the mouse and human sequences were swapped in their response (I suppose the latter is by mistake).

Reviewer #2

(Remarks to the Author)

I am OK with the revisions.

Version 3:

Reviewer comments:

Reviewer #1

(Remarks to the Author)

1. The authors could mention in the Introduction section that FXR1 gene resides on 3q26; now they describe both this region and FXR1 gene but do not explicitly connect them. They could also cite and mention previous studies' estimates on 3q26 amplification prevalence in HGSC.

2. The authors have further modified their analyses of epithelial content and tumor purity. However, they should not indicate Epcam⁺ cells as 'Tumor cells' as they do in Suppl Fig 7; this is contradictory to described results of CopyKAT analysis suggesting that there are also non-cancerous epithelial cells.

Otherwise, their updated analyses suggest that there are still aneuploid cells left after siFXR1 and thus the term "tumor microenvironment" can be used.

3. Fig. 4f and g; the UMAPs, or their legends do not indicate which cells are from siCtrl and which from siFXR.

4. Also chemotherapy modifies HGSC immune TME to increased T cell infiltration and macrophage polarization shift from M2 to M1 like (e.g. DOI: 10.1016/j.ccell.2024.11.005), suggesting that therapies that weaken cancer cells may in general also relieve cancer-inflicted immune suppression. Therefore, one cannot separate whether the effects on the immune system are specific to siFXR1, or a more general consequence of reduced tumor burden and tumor cell fitness. Please mention this

in the Discussion section when describing the siFXR associated immune changes.

5. There are still grammatical errors / broken sentences in the manuscript, please proof-read and correct.

Reviewer's Comments:

Reviewer #1 (Remarks to the Author)

The authors demonstrated that FXR1 is a promising therapeutic target for the elimination of ovarian cancer cells. They showed that siRNA-mediated silencing of FXR1 selectively induces apoptosis and G1 phase cell cycle arrest in ovarian cancer cells, but not in normal ovarian surface epithelial (OSE) and fallopian tube epithelial (FTE) cells—likely due to the relatively low expression of FXR1 in these normal cells. To enhance therapeutic efficacy, the authors incorporated locked nucleic acid (LNA) modifications and polyethylenimine (PEI) encapsulation of the siRNAs, which improved their stability, cellular and tissue uptake, and overall in vivo therapeutic performance. Finally, using single-cell RNA sequencing (scRNA-seq), they demonstrated that delivery of LNA-modified FXR1 siRNAs in mice triggered an anti-tumor immune response, characterized by tumor-associated macrophage (TAM) polarization and increased infiltration of dendritic cells (DCs), natural killer (NK) cells, and T cells.

Although the authors have performed extensive analyses of the effects of siFXR1 both in vitro and in vivo, the storyline is difficult to follow as it drowns to the vast amount of oftentimes unjustified or redundant analyses shown in 9 main and 10 supplementary figures. In addition, the choice of methods and models are not consistently justified, statistics are lacking, claims are not always supported by the data shown, and the novelty of results shown in first figures is limited. Hence, we do not recommend the manuscript for publication in Nature Communications. We delineate our criticism below and hope the authors can use these to improve their manuscript for following submission(s) to another journal.

1. Introduction does not sufficiently refer to previous work on using siRNA modifications in cancer research, including in vivo models and even clinical trials (see e.g., reviews doi.org/10.1038/s41392-020-0207-x, doi.org/10.1016/j.jconrel.2023.07.054). This leads to overstatement of the novelty of the conceptual idea behind this article.

Response: Suggested articles are now included in the introduction. A total of twelve key references (Ref. 12–23, 58) have now been incorporated into the Introduction and Discussion sections to highlight foundational studies demonstrating the therapeutic potential of siRNAs.

However, none of these prior studies have explored targeting FXR1 or other RNA-binding proteins (RBPs), likely reflecting the longstanding belief that RBPs are undruggable entities. The central objective of our study is to challenge this paradigm by demonstrating that targeting RBPs; specifically, FXR1 in tumor tissues represents a viable and innovative therapeutic strategy. Given the current lack of available small molecule inhibitors for FXR1, we employed a locked nucleic acid (LNA)-modified siRNA to achieve target-specific and efficient silencing of FXR1. Our data reveal that FXR1 knockdown not only suppresses tumor growth but also reprograms the tumor microenvironment by enhancing macrophage and immune cell populations with anti-tumor characteristics. Unlike prior studies, our work integrates single-cell transcriptomic analysis to elucidate the in vivo cellular responses to RNA interference at a single-cell resolution. By addressing this critical gap, our study provides a comprehensive

framework for advancing chemically modified FXR1-targeting siRNAs as novel therapeutics for cancer.

We have now highlighted the above description in the introduction and discussion in the revised manuscript.

2. The part of FXR1 siRNA selectively inducing apoptosis and G1 phase cell cycle arrest in ovarian cancer cells is clear, but the novelty is compromised since the authors have reported similar phenotypes previously (ref 9).

Response: We appreciate the reviewer's comment highlights our previous work on the phenotypic effects of FXR1 silencing. However, we haven't evaluated the effects of LNA-modified siRNAs in comparison with native form of siRNA in our previous work in consideration with RNA stability, and its impact on the cells in tumor microenvironment *in vivo*. Indeed, we observed that the LNA form of FXR1-siRNAs are highly stable in both *in vitro* and *in vivo* experimental settings.

In agreement with reviewer's comment, we have moved such preliminary data to revised **Supplementary Fig 1**.

3. As stated above, the current form of nine figures is laborious for the reader and way too lengthy for the relatively simple message of the manuscript. The core data of Figures 1–3 on the optimization of the siRNA could be combined to one figure, and the scRNA-seq analysis in Figures 5–9 to one to two figures.

Response: To address this comment, we have condensed our data into 6 main figures as below.

- The core data from the original Figures 1 and 2 have been combined into a single revised figure, and now presented as **Figure 1**.
- siRNA stability and siRNA uptake data presented as **Figure 2**.
- The *in vivo* results demonstrating FXR1 inhibition via LNA-formulated siRNA are presented in **Figure 3**. In this figure, we included the therapeutic effects of the LNA-modified siRNA targeting FXR1 (siFXR1) in two independent *in vivo* models: (1) intraperitoneal injection of siFXR1 in mice bearing intraperitoneally implanted OVCAR8 ovarian cancer cells, and (2) systemic administration in mice orthotopically implanted with the aggressive MDAMB231 triple-negative breast cancer (TNBC) cells in the mammary fat pad. We leveraged the TNBC model because of its common genomic features with high-Grade Serous Ovarian Carcinoma (HGSOC) such as frequent p53 and BRCA mutations and high level of genomic instability. We have also considered the advantage of measuring tumor volume in this orthotopic model while the siRNA was delivered systemically *via* retro-orbital injections.
- Overall analysis of single cell transcriptome profiling demonstrated an anti-tumor immune response upon LNA-modified FXR1 siRNA delivery, along with cell chat analysis shows cellular communications and copy number variations (CNV) as **Figure 4** in the revised manuscript.

- Sc-RNA seq data of macrophages and dendritic cell populations are now combined as a single figure and presented as **Figure 5** in the revised manuscript.
- Sc-RNA seq data of T-cells and NK cells are now combined as a single figure and presented as **Figure 6** in the revised manuscript.

4. The advantages of LNA modification and PEI encapsulation of siRNA appear

Figure-1: **a**, Oligonucleotide sequences are shown for control siRNA (siCont), two mutated siRNAs (seq2 and seq3) with mutations indicated in red both sense and antisense strands, and siRNAs targeting FXR1. **b**, Western blots to show knockdown efficiency of indicated siRNAs after 48h transfection against FXR1 in OVCAR8 ovarian cancer cells. **c**, Schema shows the schedule of injections of PEI-incorporated siCont-LNA, mutsiFXR1, native siRNA, and siFXR1-LNA injected (once/week) intraperitoneally (IP) in OVCAR8 tumor-bearing female athymic nude mice. Each group contains eight mice (n = 8). On day 45, all mice were euthanized, and tumors were collected for further analysis. **d**, Mice from a were imaged using an IVIS imager and representative images of two mice per group were presented at the indicated time point. **e**, Bioluminescent signals were from **d** and quantitated at the indicated time points and presented. **f**, Representative image (top) of the anatomy of peritoneal cavity of mice from each group. Areas circled in blue indicate tumor nodule formed in each peritoneal organ. Primary and disseminated tumors were collected from **d** and total tumor weight was recorded (bottom). Significance was determined by Student's t-test, where **p<0.01, ***p<0.001, ****p<0.0001, ns, non-significant.

promising, particularly in enhancing stability and delivery. However, the in vivo results were not fully convincing, as the native FXR1 siRNA alone already led to a significant reduction in tumor burden, making it difficult to clearly attribute additional therapeutic benefit to the modifications.

12. For the in vivo experiments, it would be helpful to test a less frequent dosing schedule (e.g., extended intervals between siRNA administrations) to more clearly demonstrate the advantage of using LNA-modified siRNAs over the native form. This would better reflect the improved stability and durability that LNA modifications are intended to provide.

Response to comments 4 and 12: We appreciate reviewer's comment (comment #4) and a thoughtful suggestion (comment #12) of testing less frequent dosing schedule between siRNA administrations to more clearly demonstrate the advantage of using LNA-modified siRNAs over the native form.

As suggested (comment#12), a new study was performed, where we injected native and LNA form

of siRNA once a week for six weeks (**Figure-1a -1c in the rebuttal letter**). As expected, we found a better difference in the total tumor weight when the tumor bearing mice were treated with LNA form of FXR1 siRNA over native FXR1 siRNA. Similarly, a reduction in the luminescence of overall tumor signal from LNA form of FXR1 siRNA treated group over native FXR1 siRNA was observed (**Fig-1d -1f in the rebuttal letter**).

We also used a mutated sequence of siRNA to further confirm whether the delivery of the mutated sequence of siFXR1 does not exhibit any tumor suppressive effects. Herein, we mutated eight nucleotides as highlighted in red letters in **Fig. 1a (rebuttal letter)** in siFXR1-2 and siFXR1-3 sequences. As expected, the transfection of mutated siRNAs (seq2 and seq3) did not exhibit any silencing of FXR1 in OVCAR8 cells compared and did not affect any change in the overall tumor growth (**Fig. 1b in the rebuttal letter**).

5. There are several results that lack sufficient statistical analyses, and hence conclusions presented based on them are not justified by the data. For example, Fig. 1c, and Fig. 3g and Fig. S4c lack quantification and statistical analyses; hence, the claims made on these data are not justified. Also, e.g., Figs. 5d, 8c, and 9f lack error bars and statistics (the data should contain two repeats according to Fig. 5a).

Response: As suggested, we have now included quantification data based on densitometric analyses in Fig. 1c (**New Sup Fig-1c in the manuscript**), Fig. 3g (**New Sup Fig-3b in the manuscript**), and S4c (**is now combined with New Fig-2g in the manuscript**).

We have also revised **Fig. 5d (New Fig.4c)**, **Fig. 8c (New Fig. 6c)**, and **Fig. 9f (New Fig. 6j)** and all similar panels show scRNA-seq data with appropriate statistical comparison between control and treatment group with p-values **in the manuscript**.

Ascites from two biological replicates were pooled to generate one sample, and two such pooled samples per treatment group (siCont-LNA and siFXR1-LNA) were used for single-cell RNA sequencing. Single-cell suspensions were prepared using tissue dissociation enzymes. During data analysis, we observed that one of the siFXR1-LNA samples exhibited reduced sequencing depth and a markedly lower number of detectable cells. This reduction is likely attributable to the therapeutic effect of siFXR1-LNA, which resulted in decreased tumor burden and ascites volume (**Revised Supplementary Fig. 5e, 5f in the manuscript**). To assess the treatment-associated changes in cellular composition, we statistically compared the percentage frequency of each cell type between the siCont-LNA and siFXR1-LNA groups (**Fig. 2 in the rebuttal letter**). In total, 16,851 high-quality cells were retained for downstream analysis, comprising approximately 11,621 cells from the siCont-LNA group and 5,230 cells from the siFXR1-LNA group (**Revised Supplementary Fig. 5h in the manuscript**). We have expanded our description in the results and method section for clarity in the revised manuscript.

6. The mouse models containing KRAS^{G12D} are not optimal, as KRAS mutations are extremely rare in HGSC. Also, the authors assessed two mouse cell lines, one with and one without MYC amplification, but chose the cell line without MYC amplification for in vivo experiments without any justification. Importantly, the authors use FXR1's role in MYC translation as a justification for its mechanistic importance in tumorigenesis but do not address it at all, even though they have a syngeneic pair of cell lines with and without MYC to directly do this.

Response: We respectfully disagree partially with some of the points related to this comment raised by the reviewer.

(1). There are only very few syngeneic ovarian cancer cell lines currently available that reliably establish tumors in immunocompetent mouse models. While, we have selected two of those lines such as Br-Luc (p53^{-/-}, brca^{-/-}, **Myc**) and C-11 (p53^{-/-}, **Myc**, & KRAS^{G12D}) for *in vitro* experiments, **the cell line that we selected Br-Luc line does not have KRAS^{G12D} was used for mouse model-based studies.**

(2). We also would like to draw reviewer's attention that both lines Br-Luc and C-11, we used harbor Myc amplification. More importantly, Br-Luc cell line used for mouse model exhibits MYC amplification.

The appropriate reference as below regarding MYC status has now been included in the revised manuscript.

Xing D, et al. A mouse model for the molecular characterization of brca1-associated ovarian carcinoma. Cancer Res. 2006.

(3). Unfortunately, we could not identify a syngeneic pair of line with and without MYC amplification to perform the suggested experiments in the given time. To further address this comment, we have included additional data examining MYC expression levels in MYC-amplified cell lines (Br-Luc and C-11) following siRNA-mediated FXR1 knockdown (**Revised Supplementary Fig. 5b in the manuscript**). As expected, MYC level was decreased in a direct correlative manner in both Br-Luc and C-11 cell lines.

7. Throughout the manuscript, the authors make unjustified claims based on CellChat analysis. CellChat results do not “determine the ligand-receptor interactions”; they present potential for interactions. CellChat results can neither reveal mechanisms but rather potential mechanisms.

Response: Thank you for the suggestion. We have now revised the text appropriately.

8. The use of CIBERSORT, meant for bulk RNA-seq signal deconvolution, to what authors state as “a more detailed characterization” when analysing differences in cell-type signature scores between specimens is not well justified. There are many widely used scoring methods—such as AUCell, ssGSEA, AddModuleScore—for this exact purpose.

Response: We would like to clarify that the CIBERSORT analysis was used only to identify the gene signature, which were then combined subsequently to calculate module scores using the ‘AddModuleScore’ function.

We have now revised the description in the Results section to avoid any confusion related to the previous text. Additionally, we have added a description of the AddModuleScore analysis in the Methods section.

9. The scRNA-seq analyses of Figure 6 are mostly obscure. For example, why do the authors calculate “overall tumor scores” to differentiate between tumor versus normal cells, when they have CopyKAT data (or can perform inferCNV) to robustly separate tumor cells from normal cells?

Response: To avoid confusion and redundancy, we have removed overall tumor score analysis from the current manuscript.

10. From the scRNA-seq data, the authors mentioned that there were about 3500 epithelial cells (Fig. S5h). Among them, about 3000 cells from siCont-LNA and 500 from siFXR1-LNA (Fig. 5d). But from lines 415–419, “highest numbers of aneuploid epithelial cells (5400) over normal epithelial cells (120 diploid cells)”, the cell number of epithelial cells is very unclear and does not match to Fig. 6a. And there were almost no tumor

epithelial cells at all from siFXR1-LNA group, making us to wonder the sense of CellChat analysis between normal rather than tumor epithelial cells and TME.

Response: We thank the reviewer for highlighting the discrepancy in the epithelial cell numbers in the text line number 415–419, where we stated that 5400 aneuploid vs. 120 diploid epithelial cells. This discrepancy happened due to a typographical error. This error has been corrected in the revised manuscript as 892 aneuploid cells and 806 diploid cells in the control group, whereas 2 aneuploid cells and 180 diploid cells were identified in the treatment group.

Regarding the cell chat, our goal was to identify the oncogenic signaling such as ECM-receptor signaling. In conjunction with overall results, we observed ECM-receptor signaling was in higher magnitude in the control, whereas siFXR1 treatment reduced the magnitude of ECM-receptor signaling.

Precisely we observed that collagens (encoded by Col4a1, Col4a2, Col4a5, and Col4a6), laminins (Lama3 and Lamc1), and fibronectin (Fn1) from tumor cells interact with their respective receptors in immune cells, potentially as a pro-tumorigenic mechanism in the control group. In contrast, cell-to-cell contact driven by MHC components and ligands activate T/NK cell functions and secreted signaling driven by IL-18, and TGF-B1 for immune cell activation were observed high in the siFXR1 treated group.

We have now revised the text to highlight the results that we obtained from cell chat analysis.

11. The author designed 5 siRNAs to target human FXR1, of which seq2 and seq3 were demonstrated with high efficiency. Later, the author used them (seq2 and seq3) also for targeting mouse FXR1, achieving good knockdown efficiency (Fig. S5a). However, the BLAST results seem they (seq2 and seq3) are not 100% matching to mouse FXR1. Have the authors checked the potential off-targets when using human FXR1 siRNAs for mouse FXR1?

Response: Seems there is a confusion regarding the alignment match of seq2 and seq3 with mouse FXR1 at reviewer's side.

We have already considered the sequence homology before our animal experiments to ensure that the siRNA sequence we selected are matching well with mouse FXR1 for our in vivo experiments.

Here we confirm again that Seq2 siRNA shows 90% match with mouse FXR1 sequence. More importantly, Seq3 siRNA shows 100% match with mouse FXR1 sequence (Figure- 3 in the rebuttal letter). We have used Seq3 siRNA in all our in vivo experiments, which is included in the experiment for sc-RNA sequencing (Revised Fig. 3, Fig. 4 and Supplementary Fig. 5 in the manuscript).

Reference sequence ID of mouse FXR1 gene was already described in the method section relevant to in vivo and sc-RNA seq experiments.

12. For the in vivo experiments, it would be helpful to test a less frequent dosing schedule (e.g., extended intervals between siRNA administrations) to more clearly demonstrate the advantage of using LNA-modified siRNAs over the native form. This would better reflect the improved stability and durability that LNA modifications are intended to provide.

Response: Thank you for the comment. This concern has already addressed in conjunction with comment # 4. Please see the response above and the **Figure-1 in the rebuttal letter**.

As suggested, less frequent dosing schedule of FXR1 siRNA more clearly demonstrated the advantage on tumor growth inhibition when using LNA-modified siRNAs over the native form.

13. For scRNA-seq experiments, why the author chose ascites and not the tumor mass? This should be clarified.

Response: Unlike other cancers, ovarian cancer is a peritoneally progressing cancer where the tumor cells shed into the peritoneal cavity, then circulates through ascites fluid before metastasizing peritoneally. While peritoneally progress, ovarian cancer promotes ascites formation, where the tumor cells grow as a cluster of stromal cells, and immune cells together. Therefore, cell clusters and tumor spheroids isolated from ascites fluid represent the tumors of advanced ovarian cancer. Supporting this notion, the studies listed below have also used cell clusters from ascites to characterize the transcriptomic changes at the single cell level.

(e.g., <https://www.nature.com/articles/s41591-020-0926-0>, *Nat Med* **26**, 1271–1279 (2020) and <https://www.nature.com/articles/s41591-020-0844-1> *Nat Med* **26**, 792–802 (2020)).

We have explained the rationale of selecting ascitic fluid in our single-cell analysis and included a description in the introduction and relevant sections of results and cited appropriate references in the revised manuscript.

Minor Comments

15. The manuscript requires a proper language check.

Response: Edited and revised.

16. Abstract: “In tumor microenvironment (TME), siFXR1-LNA diminished tumor cell proliferation” should read “In vivo, siFXR1-LNA diminished tumor cell proliferation ...”

Response: Corrected as suggested.

17. Abstract: “such as copy number gain, amplification, or deletion of genome.” should be corrected to “...deletion of genomic elements.”

Response: Revised as suggested.

18. The ‘native’ siRNAs (siFXR1-2, siFXR1-3) have significant effects on cell cycle in Fig. 1b but not in Fig. 2d; please explain this inconsistency.

Response: In our Fig. 1 experiments in the first version, we used two concentrations of siRNAs, which are relatively higher for our initial screening and then determined the effects of siRNAs on FXR1 protein at 2.5nM and 5nM concentrations [Please see Fig-1a in the revised manuscript and Figure-4 in the rebuttal letter]. We then continued using the lowest effective concentration 2.5nM on cell cycle phases using flow cytometry, Western blot analysis of proteins associated with G1 cell cycle phase, CDKs, and proteins associated with cell death.

In the next set of experiments, we evaluated the knockdown effects of FXR1 of selected siRNAs from first set of experiments such as sequence#2 (siFXR1-2) and sequence#3 (siFXR1-3) at the lowest possible concentration as 0.5, 1 and 2 nM and then selected 0.5 nM concentration of native form and its LNA form [Please see Fig-1e in the revised manuscript and Figure-5 in the rebuttal letter]. to determine their effects on cell cycle phases using flow cytometry, Western blot analysis of proteins associated with G1 cell cycle phase, CDKs, and proteins associated with cell death.

In brief, the changes in the effects on cell cycle phases between old Fig. 1b (**Supplementary Fig. 1b in the revised manuscript: -siRNA concentration used 2.5 nM**) and old Fig. 2d (**Supplementary Fig. 3a in the revised manuscript: -siRNA concentration used 0.5 nM**) are due to the difference in concentrations.

19. It is not clear what is the difference between Fig. 3g and Fig. S4c. Also, as referred to above, the result text states in both that tissues with tumor have more Texas Red signal from the siRNA, but the amount of tumor is not shown or quantified.

Response: We apologize for this confusion. Supplementary Fig. S4c is the control group of main Fig-3g. This experiment was performed to show the level of uptake of native and LNA form of FXR1-siRNA immediately (1 hr) after the injection of siRNA. We have now marked the tumors within yellow dotted box in both groups and moved this Fig. S4c to the main figure as **Fig. 2g** in the revised form of manuscript. Quantification for Texas red intensity is also provided as **Fig. 2h**.

20. Fig. 4g: how can the proportion of TUNEL-positive cells be over 100%? Doesn't this suggest that some of the TUNEL signal is artifactual?

Response: Sorry for the typographic error of % mark in the label. We have removed this error and revised the 'y axis' as average TUNEL+ cell number (**Supplementary Fig. 4c in the revised manuscript**).

21. Line 358–361: the authors said “the interactions were notably higher in the control group”, is that true?

Response: We have rewritten this statement to “the interactions between tumor cells to macrophages and other immune cells were decreased significantly upon the treatment with siFXR1-LNA” compared to the control for clarity.

22. Line 412: there is no “Fig. S5k”.

Response: Removed this error and revised the manuscript.

23. Fig. 5d y-axis label is marked “%”, which does not match the actual data. A similar problem appears in Fig. 8f.

Response: We revised the ‘y axis’ as either actual numbers as either frequency or average cell number (**Fig. 4c, Fig. 6f** in the revised form of manuscript).

24. Fig. S3b flow cytometry figure is not properly presented for visualization.

Response: Sorry for this error happened while the actual image was converted to pdf while submission. Figure is now reformatted for accuracy. (**Supplementary Fig. 3c** in the revised form of manuscript).

25. In Figure S7a, the x-axis labels “0, 1, 2” are unclear — could the authors clarify what these values represent?

Response: These populations are either aneuploid, diploid or not any defined group of cells (**Supplementary Fig. 6a** in the revised form of manuscript). We have included cell type information in the panel legend in the revised manuscript.

26. “Therefore, FXR1 is considered as an excellent molecular and actionable target for ovarian and other cancer therapy.” Isn’t this a bit overstatement? One would assume that excellent targets are targetable by small molecules.

Response: We have now revised this statement as ‘Our data suggest that FXR1 is a potential target for treatment for ovarian and other cancers exhibits FXR1 amplification’.

Reviewer #2 (Remarks to the Author)

This is a nicely executed study demonstrating the therapeutic potential of a locked nucleic acid (LNA)-based siRNA targeting FXR1 for ovarian cancer treatment. The engineered siFXR1-LNA showed enhanced stability against RNase degradation, improved tumor uptake, and potent suppression of FXR1 expression compared to native siRNA. Its delivery via polyethylenimine significantly reduced tumor growth and peritoneal metastasis in both nude and immunocompetent mouse models, without observable toxicity. Single-cell RNA sequencing revealed not only direct inhibition of FXR1 in cancer cells but also disruption of oncogenic translation pathways and normalizing the tumor microenvironment’s immune function. A range of complementary methods supports the authors’ conclusions, which are well-reasoned and convincing.

Critique:

Please state whether sc RNA-seq dataset is available and the data bank where it is deposited.

Response: GEO accession numbers were already provided as part of the reporting summary along with this submission as below:

All sc-RNA seq data generated for this study have been deposited in the NCBI Gene Expression Omnibus (GEO) under private accession number GSE292799 with a secure token code: mbihuummzvyrsp

Please comment on why the single-cell analysis was done with ascites and not tumors. The benefit of using ascites is less variability in cell types, but the benefit of tumors is the spatial context, which is crucial for cell signaling in immune infiltrates. A good part of the work was the description of cell-cell signaling inference by CellChat. Is it assumed that even after these immune cells shed into the ascites, they retain the transcripts of immune activation signaling?

Response: Related question was raised by Reviewer:1 as comment #13. Please see the response to the above comment for the rationale of using ascites samples for single cell RNA sequencing.

Unlike other cancers, ovarian cancer is a peritoneally progressing cancer where the tumor cells shed into the peritoneal cavity, then circulates through ascites fluid before metastasizing peritoneally. While peritoneally progress, ovarian cancer promotes ascites formation, where the tumor cells grow as a cluster of stromal cells, and immune cells together. Therefore, cell clusters and tumor spheroids isolated from ascites fluid represent the tumors of advanced ovarian cancer. Supporting this notion, the studies listed below have also used cell clusters from ascites to characterize the transcriptomic changes at the single cell level.

Other high-impact peer reviewed publications have also employed tumor cell clusters isolated from ascites for single cell RNA sequencing. We agree that looking the transcriptomic profile using sc-RNA seq or spatial transcriptomic data from tumor tissues will be complementary to the transcriptomic profiles obtained from ascites. However, such analyses are currently under development as part of our future work and such plans are explained in the discussion section of the revised manuscript.

Lane 413 and Fig. 6 a, c, and h: I am confused by the definition of tumor cells and nonmalignant epithelial cells. I understand that this distinction was made using CopyKAT CMV inference. Based on the CNV inference, most of the cells in the ascites are nonmalignant cells. What are these epithelial cells if they are not tumor cells? Mesothelial cells?

Response: We would like to clarify that in the control group, CNV inference using CopyKAT identified 892 aneuploid cells, 806 diploid cells, and 1,545 undefined cells. Notably, treatment with siFXR1-LNA resulted in a significant reduction in the number of aneuploid cells. In the siFXR1-LNA-treated group, the majority of cells were classified as either diploid or undefined. Based on their gene expression profiles, the diploid population likely comprises epithelial cells preserved diploid characteristics (e.g., expressing Epcam, Krt8, Krt18, and Krt19), or normal-like epithelial cells exhibiting a hybrid epithelial-mesenchymal phenotype as indicated by Zeb1, Fn1, and Cdh1 expression upon FXR1 silencing therapy and some mesothelial cells based Upk3b, Nkain4 expression (**Figure-6 in the rebuttal letter**).

Fig. 4h: It appears that there is more blood in FXR1-treated lungs compared to the control. Is this by chance in the selected photographs or there is actually more blood when whole lungs are examined? The three figures below could benefit from a panel showing H&E staining, which would clarify some of the questions below:

Response: The increased blood signal in the lungs of the FXR1-siRNA-treated group was not consistently observed across all samples and in all microscopic fields. Upon reviewing the full set of lung tissues from both treatment groups, we found that the appearance of increased blood was limited to certain field in the tissue section and does not a generalizable effect upon the treatment of FXR1-siRNA. We have included triplicate images from each group as below (**Figure-7 in the rebuttal letter**).

We also included H&E-stained images for **Fig. 3f**, and **Fig-6e**, in the revised manuscript.

Fig. 4f: Two of the panels stained with Ki67 seem to have high background – all of the cells are positive, which is unlikely even in very aggressive tumors.

Response: Thank you for this point. Images were now recaptured in higher magnification with low background settings and presented as **Fig. 3f** in the revised manuscript. Revised images are able to distinguish the Ki67 positive cells over Ki67 negative cells accurately.

Fig. 7g: There are two sets of staining shown for both control and FXR1-treated mice. It is unclear whether these represent two distinct tumors per treatment group. The nuclear staining does not clearly identify the tissue context—are these epithelial tumor cells or immune infiltrates? In the FXR1-treated group, the left panels appear to show necrotic debris rather than intact nuclei, while the right panels suggest an area of immune cell infiltration. In contrast, the control panels appear to represent tumor tissue. Additionally, the F4/80 staining appears nonspecific.

Response: We appreciate the reviewer's thoughtful evaluation of the immunostaining data and the opportunity to clarify these key points.

To avoid confusion, two sets of magnified high-resolution images for macrophages stained for Arg1 and Mrc1 markers are now shown for each group (siCont-LNA- and siFXR1-LNA-treated groups). These will address the concerns of necrosis, intragroup heterogeneity of immune cell infiltration.

In brief, the observed nuclear distribution primarily reflects tumor epithelial cells, with notable infiltration of M2-type macrophages based on tissue architecture and co-staining with macrophage markers in the control group. In contrast, siFXR1-LNA-treated tumors displayed uniformly distributed, intact nuclei, consistent with preserved tissue architecture and a marked reduction in M2 macrophage presence, aligning with an enhanced anti-tumor immune response.

We also acknowledge the concern regarding the specificity of F4/80 staining. To address this, we have included new images of tumor sections from both treatment groups, immunostained for F4/80, Arg1 and Mrc1, we also captured images at higher magnification to more clearly visualize macrophage infiltration. The new high-resolution images are now included in the revised manuscript as **Supplementary Fig 8e in the manuscript**.

Fig. 8e: The panels showing tumors from control mice have different hematoxylin staining from the siFXR1-LNA-treated mice. Do these sections represent tumors or necrotic cells?

Response: Here we clarify that the images shown in Fig. 8e (**Fig.6e** in the revised manuscript) are not H&E-stained sections. Those images are immunohistochemical staining for Cd4 and Cd8a in both the siCont-LNA and siFXR1-LNA treatment groups. This information was already included in the legend and text. To avoid confusion, we have included the corresponding H&E-stained images for the same tumor sections in the revised manuscript (**Fig 6e**).

The authors have improved the paper but still the issues are such that I cannot recommend the publication of the manuscript in Nature Communications.

A major weakness of the paper is the conclusions drawn from the scRNA-seq analysis. The analysis is displayed still very extensively in Figures 4 to 6, with conclusions used in Fig. 7. Yet, it is mostly irrelevant since it is comparing tumor microenvironment to a non-tumor microenvironment, as the authors state that the treated group has only 2 aneuploid (=cancer) cells. That means that based on the data shown, one cannot specify which of the observed changes in for example immune interactions are due to the FXR knockdown, and which are simply due to the loss of cancer. Here, the results are presented and concluded upon as if they would inform us of the effects of loss of FXR1 expression, which is misleading.

Overall, the effect of the modified siRNA on tumor growth is clearly shown in the mouse model used, with also convincing evidence of how the stabilizing modification improves the response when dosing is more sparse. Also the effect on Myc expression is now clear. However, these more solid aspects form just a half of the manuscript.

Response: We thank the Reviewer for all the constructive criticism and positive comments including the effects of modified siRNA on tumor growth, how the modification improves stabilization, improved therapeutic response when modified siRNA delivered less frequently *in vivo*. We also appreciate the positive comments related to previous concern related to cMyc.

We agree that the presence of only a small number of aneuploid cells in the FXR1 knockdown group limits the ability to distinguish between changes in tumor microenvironment particularly the immune interactions are either due to the effects of FXR1 knockdown or due to overall loss of cancer.

Reviewer 1 came into this conclusion primarily from the CNV analysis performed using copyKAT analysis.

CopyKAT is a computational tool that uses single-cell RNA sequencing data to infer genomic copy number changes in a tumor which may not fully accurate due to the limitations of this analysis method such as:

(i) not all cancer cells have high level aneuploid copy number events that can be used to distinguish normal and tumor cells such that the copy number gains like mild copy number events might be missed from the CopyKAT analysis.

(ii) CopyKAT is also considered more suitable for the analysis of subclones in tumors where many cells have expanded and share similar genotypes, rather than the analysis of replicating cancer cells are very low when cancer cells are treated with therapeutic agents like FXR1 siRNA in this instance.

Figure 1: **a**, Representative images of flow cytometry analysis of ascites fluid collected from control siRNA or FXR1 siRNA treated mice performed after immunostaining using Epcam⁺ and Cd45⁺ specific antibodies. Marked area shows either tumor cells or immune cells (left). Quantitative bar graph shows percentage of tumor cells (Epcam⁺) and immune cells (Cd45⁺) in the samples were collected from siCont-LNA or siFXR1-LNA treated group (right). **b**, Immunofluorescence (IF) analysis for Epcam and Cd45 on tumor tissue sections prepared from siCont-LNA or siFXR1-LNA treated group (left). Scale bar represents 50µm and 20µm. Bar graph shows quantitation of tumor cells and immune cells based on Epcam and Cd45 staining intensity on the tumor tissues were collected from siCont-LNA or siFXR1-LNA treated group (right). **c**, Violin plot shows the expression of genes indicated in total cells clusters isolated from siCont-LNA or siFXR1-LNA treated groups, p-values are calculated by Wilcoxon Rank Sum test. **d**, Quantitative bar graph of total tumor associated macrophages (F4/80⁺Cd11b⁺) and **e**, T cells (Cd3⁺) population collected from siCont-LNA or siFXR1-LNA treated groups. Error bars indicate mean ± SEM. Significance was determined by Student's t-test, where *p<0.05, **p<0.01.

confirm the data obtained from an algorithm like CopyKAT that predict CNV numbers from gene expression data particularly when the number of tumor cells are low in the samples.

To this end, we performed two independent and complementary approaches: flow cytometry of ascites samples and immunofluorescence (IF) staining on tumor tissues collected from the mice were treated with control siRNA and FXR1 siRNA using Epcam antibody to detect cancer cells and Cd45 antibody to detect immune cells **[Figure-1a, b in the rebuttal letter]**.

We selected Epcam because it is typically upregulated in tumor cells and can be easily detected using flow cytometry, whereas it is weakly expressed by normal epithelial cells (Imrich S et al, 2012, PMID: 22647938 and Keller L et al, PMID: 31225512).

In conjunction with the sc-RNA seq data **[Fig-4c in the manuscript]**, flow cytometry of control samples also showed high number of tumor cells (~18000) but less immune cells (~9000). Notably, the treatment of the samples collected from mice were treated with FXR1 siRNA reduced the number of tumor cells to ~5000 with an increase in immune cells to ~21000.

Similarly, the immunofluorescence (IF) of tumor nodules also showed high number of Epcam⁺ tumor cells and a few Cd45⁺ immune cells in the control group. In conjunction with scRNA-seq data and flow cytometry, the number of Epcam⁺ tumor cells was low in the FXR1 siRNA treated group with a high number of Cd45⁺ immune cells in the IF imaging of tumor tissues **[Figure-1b in the rebuttal letter]**.

We further corroborated this data in our scRNA-seq analysis, where we found an overall decrease EpCAM⁺ expression with a concomitant increase in Cd45⁺ cells in the FXR1 siRNA treated group compared to controls **[Figure-1c in the rebuttal letter]**.

We further validated the changes in overall macrophages and T-cell population using flow cytometry **[Figure- 1d, e in the rebuttal letter]**, which again showed a decrease in macrophage population and an increase in the overall T-cell population similarly as we identified in our scRNA-seq data **[Fig-4c in the manuscript]**.

In summary, both our flow cytometry and IF analysis indeed identified a significant number of tumor cell population along with an increase in infiltrated immune cells in the samples were collected from the mice were treated with FXR1 siRNA **[Figure-1a, b in the rebuttal letter]**. These data further suggest that the tumor-immune cell interactions that we identified are important and valid.

This data prompted us to presume that CopyKAT analysis may not be sufficient to identify all aneuploid cells when perform the analysis in default settings where the samples contains a small number of cancer cells or when the samples possess diploid-like genomes with a few copy number alterations particularly due to therapeutic interventions. Therefore, two aneuploid cells were identified in the FXR1-siRNA treated group by CopyKAT could be due to the low sensitivity of this algorithm under default settings. To address this concern, we improved CopyKAT's KS-based segmentation threshold by setting $KS.cut = 0.05$ without changing any other inputs and parameters. KS-based segmentation mainly controls the stringency of breakpoint detection to classify the cells as aneuploid based on how many CNV segments are detected in the analysis. Strikingly, such improvement in the settings estimated more aneuploid cell fraction in the sample group were treated with FXR1 siRNA without affecting the qualitative conclusion [Figure-II in the rebuttal letter].

Of note, our refined CopyKAT analysis identified **93 aneuploid cells/315 epithelial cells (~30%) in the siFXR1-LNA treated group** versus the control group where we found 1609 aneuploid cells/3243 epithelial cells (~50%). While a decrease in the aneuploid cells were observed in the siFXR1-LNA treated group; an increase in the normal like diploid cells was also observed in the siFXR1-LNA treated group compared to the control.

Taken together our flowcytometric and immunofluorescence analysis along with a refined CopyKAT analysis confirmed that there is some tumor cells present in the mice even after the FXR1 siRNA treatment. Therefore, the composition of cells in the TME and the immune cell interactions with both tumor cells and normal epithelial cells are important and biologically relevant.

Response to comment 11:

The match of seq3 is not 100% as the authors state in their response. The authors fail to highlight the “AGA” at the end of the sequence in their response, although it is present in Table S1: “Seq3: GCUUACUUGAUAAUACAGAAUCAGA”. Hence, their response is incorrect.

Response:

We apologize for the confusion and partial response in our previous rebuttal letter. We agree that one nucleotide is not match with mouse gene in terms of complementarity when consider the whole 25 nucleotides of our siRNA, which is now corrected in the result section of the manuscript (Page-14).

To confirm that the actions of siFXR1 is operated through knocking down of FXR1, we rescued FXR1 in the mouse Br-Luc cell line using wildtype FXR1 expressing plasmid and also a mutated FXR1 plasmids where four nucleotides were changed to reduce the complementarity with FXR1 siRNA that disrupts siRNA binding without affecting amino acid order. Notably, the loss of FXR1 upon FXR1 siRNA reduced cell proliferation, which was abrogated when 4 nucleotides were mutated to reduce the complementary with siFXR1 [Figure IIIa to c, rebuttal letter].

We also like to highlight that we observed a robust and reproducible knockdown of Fxr1 mRNA and protein in two different mouse ovarian cancer cells despite this single nucleotide mismatch [Figure III d , e rebuttal letter and in Sup Fig-5 and Sup Fig-7].

We also determined the levels of MYC, which we have reported as one of the main target of FXR1 (George et al, 2021; Cell Reports, PMID: 34731628) and found that FXR1 silencing using our siRNA inhibited the levels of MYC gene. It was also reported by Shoba Vasudevan and colleagues that FXR1 causes RNA nuclear protein complex formation where it represses the expression of TNF α (Shobha Vasudevan and colleagues, Cell, 2007 PMID: 17382880). In complement to this hypothesis, we found that FXR1 silencing using our siRNA upregulated the levels of TNF α in both cell lines [Figure III d, e rebuttal letter]. We also extended our analysis to a few non-targets of FXR1 based on the AUA Rich Elements (ARE) and found that none of those genes were affected by FXR1 knockdown [Figure III d, e, rebuttal letter].

Additionally, we also considered to use long nucleotide sequence siRNAs better knockdown efficiency, with less off target effects and better stability than short sequence RNAs. The standard size of an siRNA is considered about 21 nucleotides in length, with a range of 19 to 25 nucleotides in length. In this case, we used a highest possible

length of 25-base siRNA, because studies have demonstrated that siRNAs with longer nucleotide length can serve as natural substrate for the Dicer enzyme and enhanced Risc (RNA induced silencing complex) incorporation for gene silencing with less off-target effects (John Rossi and colleagues, Nature Biotechnology, 2005, PMID: 15619617).

Moreover, with their high potency, Dicer-substrate siRNAs can be used at very low concentrations, which again minimizes the potential for off-target silencing, where the siRNA inadvertently binds to and silences unintended genes that have partial sequence homology (John Rossi and colleagues, Nature Biotechnology, 2005, PMID: 15619617).

Considering all of these points, we used very low siRNA concentrations in our in vitro experiments in both human and mouse cell lines (0.5 nM to 2 nM) to minimize off-target effects. Previous studies have also shown that lower siRNA doses reduce off-target silencing of non-complementary transcripts, including the inadvertent induction of immune-response genes (PMID: 21750714, 16682561).

In our in vivo studies, we further limited the siRNA dose to 500 µg/kg body weight, whereas most studies from industrial groups developing RNAi therapeutics for clinical trials typically use 10–50 mg/kg body weight in their preclinical models (PMID: 38549376, 40349108).

Taken together, we have carefully implemented multiple precautions to minimize off-target effects in both human and mouse models.

Minor:

Response to comment 6.

The authors now state that the model used, Br-Luc does not have KRAS mutation, even though their previous manuscript version that was reviewed stated on rows 311-312: Br-Luc (genotype: p53^{-/-}; BRCA1^{-/-}; myc; KrasG12D; Akt-myr) and C11 (genotype: p53^{-/-}; BRCA1^{-/-}; myc; KrasG12D). If there was an error in this version the authors could have simply acknowledged that there was an error.

Response: We acknowledge this as an error happened during the first submission. The correct genotype of the Br-Luc model is p53^{-/-}; brca1^{-/-}; myc; Akt-myr and does not exhibit KrasG12D. This has been corrected in the revised manuscript. We apologize for the confusion and appreciate the opportunity to clarify this point.

Response to comment 11:

In addition to the misleading highlighting of the target sequence, the mouse and human sequences were swapped in their response (I suppose the latter is by mistake).

Response: We apologize and thank you to the Reviewer for finding this error, which is corrected in the revised manuscript and presented as **Supplementary Figure 5A**.

Reviewer #2 (Remarks to the Author):

I am OK with the revisions.

We sincerely thank **Reviewer #2** for their time, thoughtful evaluation, and positive feedback.

We sincerely hope both reviewers and the Editors will find the revisions are satisfactory and this manuscript is ready for acceptance subsequently.

REVIEWERS' COMMENTS

Reviewer #1 (Remarks to the Author):

The authors could mention in the Introduction section that FXR1 gene resides on 3q26; now they describe both this region and FXR1 gene but do not explicitly connect them. They could also cite and mention previous studies' estimates on 3q26 amplification prevalence in HGSC.

Response: Revised as suggested and included appropriate references.

The authors have further modified their analyses of epithelial content and tumor purity. However, they should not indicate Epcam+ cells as 'Tumor cells' as they do in Suppl Fig 7; this is contradictory to described results of CopyKAT analysis suggesting that there are also non-cancerous epithelial cells.

Otherwise, their updated analyses suggest that there are still aneuploid cells left after siFXR1 and thus the term "tumor microenvironment" can be used.

Response: We have revised the label to Epcam+ cells in Suppl Fig 7 and revised the corresponding description appropriately in the result section.

3. Fig. 4f and g; the UMAPs, or their legends do not indicate which cells are from siCtrl and which from siFXR.

Response: Revised the legends for the main figures 4f and 4g as suggested. To clarify further, two new supplementary panels with color code legends are included as Supplementary Figure 7a and 7b in the revised manuscript.

4. Also chemotherapy modifies HGSC immune TME to increased T cell infiltration and macrophage polarization shift from M2 to M1 like (e.g. DOI: 10.1016/j.ccell.2024.11.005), suggesting that therapies that weaken cancer cells may in general also relieve cancer-inflicted immune suppression. Therefore, one cannot separate whether the effects on the immune system are specific to siFXR1, or a more general consequence of reduced tumor burden and tumor cell fitness. Please mention this in the Discussion section when describing the siFXR associated immune changes.

Response: Thanks for the suggestion. We have now included this point in the discussion section and cited the mentioned manuscript respectively.

5. There are still grammatical errors / broken sentences in the manuscript, please proof-read and correct.

Response: Manuscript has been proof-read and revised accordingly.